# UNDERSTANDING APPROXIMATE AND UNROLLED DICTIONARY LEARNING FOR PATTERN RECOVERY

**Benoît Malézieux**
Université Paris-Saclay, Inria, CEA
L2S, Université Paris-Saclay–CNRS–CentraleSupelec
benoit.malezieux@inria.fr

**Thomas Moreau**
Université Paris-Saclay, Inria, CEA
Palaiseau, 91120, France
thomas.moreau@inria.fr

**Matthieu Kowalski**
L2S, Université Paris-Saclay–CNRS–CentraleSupelec
Gif-sur-Yvette, 91190, France
matthieu.kowalski@universite-paris-saclay.fr

## ABSTRACT

Dictionary learning consists of finding a sparse representation from noisy data and is a common way to encode data-driven prior knowledge on signals. Alternating minimization (AM) is standard for the underlying optimization, where gradient descent steps alternate with sparse coding procedures. The major drawback of this method is its prohibitive computational cost, making it unpractical on large real-world data sets. This work studies an approximate formulation of dictionary learning based on unrolling and compares it to alternating minimization to find the best trade-off between speed and precision. We analyze the asymptotic behavior and convergence rate of gradients estimates in both methods. We show that unrolling performs better on the support of the inner problem solution and during the first iterations. Finally, we apply unrolling on pattern learning in magnetoencephalography (MEG) with the help of a stochastic algorithm and compare the performance to a state-of-the-art method.

## 1 INTRODUCTION

Pattern learning provides insightful information on the data in various biomedical applications. Typical examples include the study of magnetoencephalography (MEG) recordings, where one aims to analyze the electrical activity in the brain from measurements of the magnetic field around the scalp of the patient (Dupré la Tour et al., 2018). One may also mention neural oscillations study in the local field potential (Cole & Voytek, 2017) or QRS complex detection in electrocardiograms (Xiang et al., 2018) among others.

Dictionary learning (Olshausen & Field, 1997; Aharon et al., 2006; Mairal et al., 2009) is particularly efficient on pattern learning tasks, such as blood cells detection (Yellin et al., 2017) and MEG signals analysis (Dupré la Tour et al., 2018). This framework assumes that the signal can be decomposed into a sparse representation in a redundant basis of patterns – also called atoms. In other words, the goal is to recover a sparse code $Z \in \mathbb{R}^{n \times T}$ and a dictionary $D \in \mathbb{R}^{m \times n}$ from noisy measurements $Y \in \mathbb{R}^{m \times T}$ which are obtained as the linear transformation $DZ$, corrupted with noise $B \in \mathbb{R}^{m \times T}$: $Y = DZ + B$. Theoretical elements on identifiability and local convergence have been proven in several studies (Gribonval et al., 2015; Haeffele & Vidal, 2015; Agarwal et al., 2016; Sun et al., 2016). Sparsity-based optimization problems related to dictionary learning generally rely on the usage of the $\ell_0$ or $\ell_1$ regularizations. In this paper, we study Lasso-based (Tibshirani, 1996) dictionary learning where the dictionary $D$ is learned in a set of constraints $\mathcal{C}$ by solving

$$\min_{Z \in \mathbb{R}^{n \times T}, D \in \mathcal{C}} F(Z, D) \triangleq \frac{1}{2} \|DZ - Y\|_2^2 + \lambda \|Z\|_1 \quad . \tag{1}$$

Dictionary learning can be written as a bi-level optimization problem to minimize the cost function with respect to the dictionary only, as mentioned in Mairal et al. (2009),

$$\min_{\boldsymbol{D} \in \mathcal{C}} G(\boldsymbol{D}) \triangleq F(\boldsymbol{Z}^*(\boldsymbol{D}), \boldsymbol{D}) \quad \text{with} \quad \boldsymbol{Z}^*(\boldsymbol{D}) = \arg\min_{\boldsymbol{Z} \in \mathbb{R}^{n \times T}} F(\boldsymbol{Z}, \boldsymbol{D}) \ . \tag{2}$$

Computing the data representation $\boldsymbol{Z}^*(\boldsymbol{D})$ is often referred to as the inner problem, while the global minimization is the outer problem. Classical constraint sets include the unit norm, where each atom is normalized to avoid scale-invariant issues, and normalized convolutional kernels to perform Convolutional Dictionary Learning (Grosse et al., 2007).

Classical dictionary learning methods solve this bi-convex optimization problem through *Alternating Minimization* (AM) (Mairal et al., 2009). It consists in minimizing the cost function $F$ over $\boldsymbol{Z}$ with a fixed dictionary $\boldsymbol{D}$ and then performing projected gradient descent to optimize the dictionary with a fixed $\boldsymbol{Z}$. While AM provides a simple strategy to perform dictionary learning, it can be inefficient on large-scale data sets due to the need to resolve the inner problems precisely for all samples. In recent years, many studies have focused on algorithm unrolling (Tolooshams et al., 2020; Scetbon et al., 2021) to overcome this issue. The core idea consists of unrolling the algorithm, which solves the inner problem, and then computing the gradient with respect to the dictionary with the help of back-propagation through the iterates of this algorithm. Gregor & LeCun (2010) popularized this method and first proposed to unroll ISTA (Daubechies et al., 2004) – a proximal gradient descent algorithm designed for the Lasso – to speed up the computation of $\boldsymbol{Z}^*(\boldsymbol{D})$. The $N + 1$-th layer of this network – called LISTA – is obtained as $\boldsymbol{Z}_{N+1} = ST_{\frac{\lambda}{L}}(\boldsymbol{W}^1 \boldsymbol{Y} + \boldsymbol{W}^2 \boldsymbol{Z}_N)$, with $ST$ being the soft-thresholding operator. This work has led to many contributions aiming at improving this method and providing theoretical justifications in a supervised (Chen et al., 2018; Liu & Chen, 2019) or unsupervised (Moreau & Bruna, 2017; Ablin et al., 2019) setting. For such unrolled algorithms, the weights $\boldsymbol{W}^1$ and $\boldsymbol{W}^2$ can be re-parameterized as functions of $\boldsymbol{D}$ – as illustrated in Figure A in appendix – such that the output $\boldsymbol{Z}_N(\boldsymbol{D})$ matches the result of $N$ iterations of ISTA, *i.e.*

$$\boldsymbol{W}_D^1 = \frac{1}{L} \boldsymbol{D}^\top \quad \text{and} \quad \boldsymbol{W}_D^2 = \left( \boldsymbol{I} - \frac{1}{L} \boldsymbol{D}^\top \boldsymbol{D} \right), \quad \text{where} \quad L = \|\boldsymbol{D}\|^2 \ . \tag{3}$$

Then, the dictionary can be learned by minimizing the loss $F(\boldsymbol{Z}_N(\boldsymbol{D}), \boldsymbol{D})$ over $\boldsymbol{D}$ with back-propagation. This approach is generally referred to as *Deep Dictionary Learning* (DDL). DDL and variants with different kinds of regularization (Tolooshams et al., 2020; Lecouat et al., 2020; Scetbon et al., 2021), image processing based on metric learning (Tang et al., 2020), and classification tasks with scattering (Zarka et al., 2019) have been proposed in the literature, among others. While these techniques have achieved good performance levels on several signal processing tasks, the reasons they speed up the learning process are still unclear.

In this work, we study unrolling in Lasso-based dictionary learning as an approximate bi-level optimization problem. What makes this work different from Bertrand et al. (2020), Ablin et al. (2020) and Tolooshams & Ba (2021) is that we study the instability of non-smooth bi-level optimization and unrolled sparse coding out of the support, which is of major interest in practice with a small number of layers. In Section 2, we analyze the convergence of the Jacobian computed with automatic differentiation and find out that its stability is guaranteed on the support of the sparse codes only. De facto, numerical instabilities in its estimation make unrolling inefficient after a few dozen iterations. In Section 3, we empirically show that unrolling leads to better results than AM only with a small number of iterations of sparse coding, making it possible to learn a good dictionary in this setting. Then we adapt a stochastic approach to make this method usable on large data sets, and we apply it to pattern learning in magnetoencephalography (MEG) in Section 4. We do so by adapting unrolling to rank one convolutional dictionary learning on multivariate time series (Dupré la Tour et al., 2018). We show that there is no need to unroll more than a few dozen iterations to obtain satisfying results, leading to a significant gain of time compared to a state-of-the-art algorithm.

## 2 BI-LEVEL OPTIMIZATION FOR APPROXIMATE DICTIONARY LEARNING

As $\boldsymbol{Z}^*(\boldsymbol{D})$ does not have a closed-form expression, $G$ cannot be computed directly. A solution is to replace the inner problem $\boldsymbol{Z}^*(\boldsymbol{D})$ by an approximation $\boldsymbol{Z}_N(\boldsymbol{D})$ obtained through $N$ iterations of a numerical optimization algorithm or its unrolled version. This reduces the problem to minimizing $G_N(\boldsymbol{D}) \triangleq F(\boldsymbol{Z}_N(\boldsymbol{D}), \boldsymbol{D})$. The first question is how sub-optimal global solutions of $G_N$ are

compared to the ones of $G$. Proposition 2.1 shows that the global minima of $G_N$ converge as fast as the numerical approximation $\boldsymbol{Z}_N$ in function value.

**Proposition 2.1** *Let $\boldsymbol{D}^* = \arg\min_{\boldsymbol{D} \in \mathcal{C}} G(\boldsymbol{D})$ and $\boldsymbol{D}_N^* = \arg\min_{\boldsymbol{D} \in \mathcal{C}} G_N(\boldsymbol{D})$, where $N$ is the number of unrolled iterations. We denote by $K(\boldsymbol{D}^*)$ a constant depending on $\boldsymbol{D}^*$, and by $C(N)$ the convergence speed of the algorithm, which approximates the inner problem solution. We have*

$$G_N(\boldsymbol{D}_N^*) - G(\boldsymbol{D}^*) \leq K(\boldsymbol{D}^*) C(N) \ .$$

The proofs of all theoretical results are deferred to Appendix C. Proposition 2.1 implies that when $\boldsymbol{Z}_N$ is computed with FISTA (Beck & Teboulle, 2009), the function value for global minima of $G_N$ converges with speed $C(N) = \frac{1}{N^2}$ towards the value of the global minima of $F$. Therefore, solving the inner problem approximately leads to suitable solutions for equation 2, given that the optimization procedure is efficient enough to find a proper minimum of $G_N$. As the computational cost of $z_N$ increases with $N$, the choice of $N$ results in a trade-off between the precision of the solution and the computational efficiency, which is critical for processing large data sets.

Moreover, learning the dictionary and computing the sparse codes are two different tasks. The loss $G_N$ takes into account the dictionary and the corresponding approximation $\boldsymbol{Z}_N(\boldsymbol{D})$ to evaluate the quality of the solution. However, the dictionary evaluation should reflect its ability to generate the same signals as the ground truth data and not consider an approximate sparse code that can be recomputed afterward. Therefore, we should distinguish the ability of the algorithm to recover a good dictionary from its ability to learn the dictionary and the sparse codes at the same time. In this work, we use the metric proposed in Moreau & Gramfort (2020) for convolutions to evaluate the quality of the dictionary. We compare the atoms using their correlation and denote as $C$ the cost matrix whose entry $i, j$ compare the atom $i$ of the first dictionary and $j$ of the second. We define a sign and permutation invariant metric $S(C) = \max_{\sigma \in \mathfrak{S}_n} \frac{1}{n} \sum_{i=1}^n |C_{\sigma(i),i}|$, where $\mathfrak{S}_n$ is the group of permutations of $[1, n]$. This metric corresponds to the best linear sum assignment on the cost matrix $C$, and it can be computed with the Hungarian algorithm. Note that doing so has several limitations and that evaluating the dictionary is still an open problem. Without loss of generality, let $T = 1$ and thus $\boldsymbol{z} \in \mathbb{R}^n$ in the rest of this section.

**Gradient estimation in dictionary learning.** Approximate dictionary learning is a non-convex problem, meaning that good or poor local minima of $G_N$ may be reached depending on the initialization, the optimization path, and the structure of the problem. Therefore, a gradient descent on $G_N$ has no guarantee to find an adequate minimizer of $G$. While complete theoretical analysis of these problems is arduous, we propose to study the correlation between the gradient obtained with $G_N$ and the actual gradient of $G$, as a way to ensure that the optimization dynamics are similar. Once $\boldsymbol{z}^*(D)$ is known, Danskin (1967, Thm 1) states that $g^*(\boldsymbol{D}) = \nabla G(\boldsymbol{D})$ is equal to $\nabla_2 F(\boldsymbol{z}^*(\boldsymbol{D}), \boldsymbol{D})$, where $\nabla_2$ indicates that the gradient is computed relatively to the second variable in $F$. Even though the inner problem is non-smooth, this result holds as long as the solution $\boldsymbol{z}^*(\boldsymbol{D})$ is unique. In the following, we will assume that $\boldsymbol{D}^\top \boldsymbol{D}$ is invertible on the support of $\boldsymbol{z}^*(\boldsymbol{D})$, which implies the uniqueness of $\boldsymbol{z}^*(\boldsymbol{D})$. This occurs with probability one if D is sampled from a continuous distribution (Tibshirani, 2013). AM and DDL differ in how they estimate the gradient of $G$. AM relies on the analytical formula of $g^*$ and uses an approximation $\boldsymbol{z}_N$ of $\boldsymbol{z}^*$, leading to the approximate gradient $g_N^1(\boldsymbol{D}) = \nabla_2 F(\boldsymbol{z}_N(\boldsymbol{D}), \boldsymbol{D})$. We evaluate how well $g_N^1$ approximates $g^*$ in Proposition 2.2.

**Proposition 2.2** *Let $\boldsymbol{D} \in \mathbb{R}^{m \times n}$. Then, there exists a constant $L_1 > 0$ such that for every number of iterations $N$*

$$\left\| g_N^1 - g^* \right\| \leq L_1 \left\| \boldsymbol{z}_N(\boldsymbol{D}) - \boldsymbol{z}^*(\boldsymbol{D}) \right\| \ .$$

Proposition 2.2 shows that $g_N^1$ converges as fast as the iterates of ISTA converge. DDL computes the gradient automatically through $\boldsymbol{z}_N(\boldsymbol{D})$. As opposed to AM, this directly minimizes the loss $G_N(\boldsymbol{D})$. Automatic differentiation yields a sub-gradient $g_N^2(\boldsymbol{D})$ such that

$$g_N^2(\boldsymbol{D}) \in \nabla_2 F(\boldsymbol{z}_N(\boldsymbol{D}), \boldsymbol{D}) + \mathbf{J}_N^+ \left( \partial_1 F(\boldsymbol{z}_N(\boldsymbol{D}), \boldsymbol{D}) \right) \ , \tag{4}$$

where $\mathbf{J}_N : \mathbb{R}^{m \times n} \to \mathbb{R}^n$ is the weak Jacobian of $\boldsymbol{z}_N(\boldsymbol{D})$ with respect to $\boldsymbol{D}$ and $\mathbf{J}_N^+$ denotes its adjoint. The product between $\mathbf{J}_N^+$ and $\partial_1 F(\boldsymbol{z}_N(\boldsymbol{D}), \boldsymbol{D})$ is computed via automatic differentiation.

**Proposition 2.3** *Let $\boldsymbol{D} \in \mathbb{R}^{m \times n}$. Let $S^*$ be the support of $\boldsymbol{z}^*(\boldsymbol{D})$, $S_N$ be the support of $\boldsymbol{z}_N$ and $\widetilde{S}_N = S_N \cup S^*$. Let $f(\boldsymbol{z}, \boldsymbol{D}) = \frac{1}{2} \|\boldsymbol{D}\boldsymbol{z} - \boldsymbol{y}\|_2^2$ be the data-fitting term in $F$. Let $R(\mathbf{J}, \widetilde{S}) = \mathbf{J}^+\big(\nabla_{1,1}^2 f(\boldsymbol{z}^*, \boldsymbol{D}) \odot \mathbb{1}_{\widetilde{S}}\big) + \nabla_{2,1}^2 f(\boldsymbol{z}^*, \boldsymbol{D}) \odot \mathbb{1}_{\widetilde{S}}$. Then there exists a constant $L_2 > 0$ and a sub-sequence of (F)ISTA iterates $\boldsymbol{z}_{\phi(N)}$ such that for all $N \in \mathbb{N}$:*

$$\exists\, \boldsymbol{g}_{\phi(N)}^2 \in \nabla_2 f(\boldsymbol{z}_{\phi(N)}, \boldsymbol{D}) + \mathbf{J}_{\phi(N)}^+\Big(\nabla_1 f(\boldsymbol{z}_{\phi(N)}, \boldsymbol{D}) + \lambda \partial_{\|\cdot\|_1}(\boldsymbol{z}_{\phi(N)})\Big) \text{ s.t. :}$$

$$\left\|\boldsymbol{g}_{\phi(N)}^2 - \boldsymbol{g}^*\right\| \leq \left\|R(\mathbf{J}_{\phi(N)}, \widetilde{S}_{\phi(N)})\right\| \left\|\boldsymbol{z}_{\phi(N)} - \boldsymbol{z}^*\right\| + \frac{L_2}{2} \left\|\boldsymbol{z}_{\phi(N)} - z^*\right\|^2 \quad.$$

*This sub-sequence $\boldsymbol{z}_{\phi(N)}$ corresponds to iterates on the support of $\boldsymbol{z}^*$.*

Proposition 2.3 shows that $\boldsymbol{g}_N^2$ may converge faster than $\boldsymbol{g}_N^1$ once the support is reached.

Ablin et al. (2020) and Tolooshams & Ba (2021) have studied the behavior of strongly convex functions, as it is the case on the support, and found similar results. This allowed Tolooshams & Ba (2021) to focus on support identification and show that automatic differentiation leads to a better gradient estimation in dictionary learning on the support under minor assumptions.

However, we are also interested in characterizing the behavior outside of the support, where the gradient estimation is difficult because of the sub-differential. In practice, automatic differentiation uses the sign operator as a sub-gradient of $\|\cdot\|_1$. The convergence behavior of $\boldsymbol{g}_N^2$ is also driven by $R(\mathbf{J}_N, \widetilde{S}_N)$ and thus by the weak Jacobian computed via back-propagation. We first compute a closed-form expression of the weak Jacobian of $\boldsymbol{z}^*(\boldsymbol{D})$ and $\boldsymbol{z}_N(\boldsymbol{D})$. We then show that $R(\mathbf{J}_N, \widetilde{S}_N) \leq L \|\mathbf{J}_N - \mathbf{J}^*\|$ and we analyze the convergence of $\mathbf{J}_N$ towards $\mathbf{J}^*$.

**Study of the Jacobian.** The computation of the Jacobian can be done by differentiating through ISTA. In Theorem 2.4, we show that $\mathbf{J}_{N+1}$ depends on $\mathbf{J}_N$ and the past iterate $z_N$, and converges towards a fixed point. This formula can be used to compute the Jacobian during the forward pass, avoiding the computational cost of back-propagation and saving memory.

**Theorem 2.4** *At iteration $N + 1$ of ISTA, the weak Jacobian of $\boldsymbol{z}_{N+1}$ relatively to $D_l$, where $D_l$ is the $l$-th row of $\boldsymbol{D}$, is given by induction:*

$$\frac{\partial(\boldsymbol{z}_{N+1})}{\partial D_l} = \mathbb{1}_{|z_{N+1}|>0} \odot \left(\frac{\partial(\boldsymbol{z}_N)}{\partial D_l} - \frac{1}{L}\left(D_l \boldsymbol{z}_N^\top + (D_l^\top \boldsymbol{z}_N - y_l)\boldsymbol{I}_n + \boldsymbol{D}^\top \boldsymbol{D} \frac{\partial(\boldsymbol{z}_N)}{\partial D_l}\right)\right) \quad.$$

$\frac{\partial(\boldsymbol{z}_N)}{\partial D_l}$ *will be denoted by $J_l^N$. It converges towards the weak Jacobian $J_l^*$ of $\boldsymbol{z}^*$ relatively to $D_l$, whose values are*

$$J_{l\ S^*}^* = -(D_{:,S^*}^\top D_{:,S^*})^{-1}(D_l \boldsymbol{z}^{*\top} + (D_l^\top \boldsymbol{z}^* - y_l)\boldsymbol{I}_n)_{S^*} \quad,$$

*on the support $S^*$ of $\boldsymbol{z}^*$, and 0 elsewhere. Moreover, $R(\mathbf{J}^*, S^*) = 0$.*

This result is similar to Bertrand et al. (2020) where the Jacobian of $z$ is computed over $\lambda$ to perform hyper-parameter optimization in Lasso-type models. Using $R(\mathbf{J}^*, S^*) = 0$, we can write

$$\left\|R(\mathbf{J}_N, \widetilde{S}_N)\right\| \leq \left\|R(\mathbf{J}_N, \widetilde{S}_N) - R(\mathbf{J}^*, S^*)\right\| \leq L\|\mathbf{J}_N - \mathbf{J}^*\| \quad, \tag{5}$$

as $\left\|\nabla_{1,1}^2 f(z^*, D)\right\|_2 = L$. If the back-propagation were to output an accurate estimate $\mathbf{J}_N$ of the weak Jacobian $\mathbf{J}^*$, $\left\|R(\mathbf{J}_N, \widetilde{S}_N)\right\|$ would be 0, and the convergence rate of $\boldsymbol{g}_N^2$ could be twice as fast as the one of $\boldsymbol{g}_N^1$. To quantify this, we now analyze the convergence of $\mathbf{J}_N$ towards $\mathbf{J}^*$. In Proposition 2.5, we compute an upper bound of $\left\|J_l^N - J_l^*\right\|$ with possible usage of truncated back-propagation (Shaban et al., 2019). Truncated back-propagation of depth $K$ corresponds to an initial estimate of the Jacobian $\mathbf{J}_{N-K} = 0$ and iterating the induction in Theorem 2.4.

**Proposition 2.5** *Let $N$ be the number of iterations and $K$ be the back-propagation depth. We assume that $\forall n \geq N-K$, $S^* \subset S_n$. Let $\bar{E}_N = S_n \setminus S^*$, let $L$ be the largest eigenvalue of $D_{:,S^*}^\top D_{:,S^*}$, and let $\mu_n$ be the smallest eigenvalue of $D_{:,S_n}^\top D_{:,S_{n-1}}$. Let $B_n = \left\|P_{\bar{E}_n} - D_{:,\bar{E}_n}^\top D_{:,S^*}^{\dagger\top} P_{S^*}\right\|$, where $P_S$ is the projection on $\mathbb{R}^S$ and $\boldsymbol{D}^\dagger$ is the pseudo-inverse of $\boldsymbol{D}$. We have*

$$\left\|J_l^N - J_l^*\right\| \leq \prod_{k=1}^{K}\left(1 - \frac{\mu_{N-k}}{L}\right)\|J_l^*\| + \frac{2}{L}\|D_l\| \sum_{k=0}^{K-1} \prod_{i=1}^{k}(1 - \frac{\mu_{N-i}}{L})\left(\left\|z_l^{N-k} - z_l^*\right\| + B_{N-k}\|z_l^*\|\right) \quad.$$

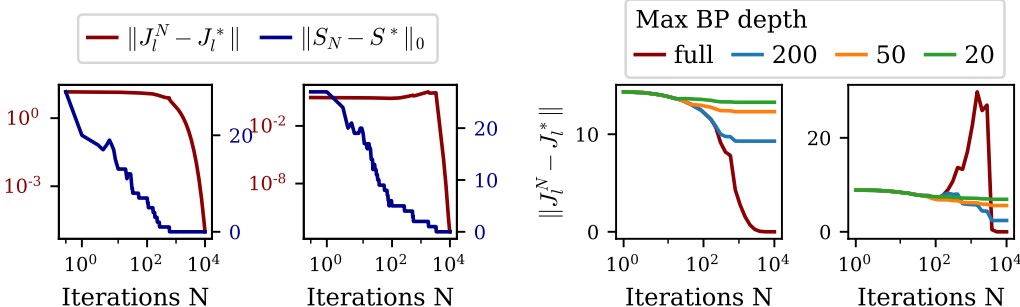

Figure 1: Average convergence of $J_l^N$ towards $J_l^*$ for two samples from the same data set, generated with a random Gaussian matrix. $\left\| J_l^* - J_l^N \right\|$ converges linearly on the support in both cases. However, for sample 2, full back-propagation makes the convergence unstable, and truncated back-propagation improves its behavior, as described in Proposition 2.5. The proportion of stable and unstable samples in this particular example is displayed in Figure 2.

Proposition 2.5 reveals multiple stages in the Jacobian estimation. First, one can see that if all iterates used for the back-propagation lie on the support $S^*$, the Jacobian estimate has a quasi-linear convergence, as shown in the following corollary.

**Corollary 2.6** *Let $\mu > 0$ be the smallest eigenvalue of $D_{:,S^*}^\top D_{:,S^*}$. Let $K \leq N$ be the back-propagation depth and let $\Delta_N = F(z_N, D) - F(z^*, D) + \frac{L}{2} \|z_N - z^*\|$. Suppose that $\forall n \in [N - K, N]$; $S_n \subset S^*$. Then, we have*

$$\left\| \boldsymbol{J}_l^* - \boldsymbol{J}_l^N \right\| \leq \left(1 - \frac{\mu}{L}\right)^K \left\| \boldsymbol{J}_l^* \right\| + K \left(1 - \frac{\mu}{L}\right)^{K-1} \|D_l\| \frac{4\Delta_{N-K}}{L^2} \ .$$

Once the support is reached, ISTA also converges with the same linear rate $(1 - \frac{\mu}{L})$. Thus the gradient estimate $\boldsymbol{g}_N^2$ converges almost twice as fast as $\boldsymbol{g}_N^1$ in the best case – with optimal subgradient – as $\mathcal{O}(K(1 - \frac{\mu}{L})^{2K})$. This is similar to Ablin et al. (2020, Proposition.5) and Tolooshams & Ba (2021). Second, Proposition 2.5 shows that $\left\| \boldsymbol{J}_l^* - \boldsymbol{J}_l^N \right\|$ may increase when the support is not well-estimated, leading to a deterioration of the gradient estimate. This is due to an accumulation of errors materialized by the sum in the right-hand side of the inequality, as the term $B_N \|\boldsymbol{z}^*\|$ may not vanish to 0 as long as $S_N \not\subset S^*$. Interestingly, once the support is reached at iteration $S < N$, the errors converge linearly towards 0, and we recover the fast estimation of $\boldsymbol{g}^*$ with $\boldsymbol{g}^2$. Therefore, Lasso-based DDL should either be used with a low number of steps or truncated back-propagation to ensure stability. These results apply for all linear dictionaries, including convolutions.

**Numerical illustrations.** We now illustrate these theoretical results depending on the number $N$ of unrolled iterations. The data are generated from a random Gaussian dictionary $\boldsymbol{D}$ of size $30 \times 50$, with Bernoulli-Gaussian sparse codes $\boldsymbol{z}$ (sparsity 0.3, $\sigma_z^2 = 1$), and Gaussian noise ($\sigma_{noise}^2 = 0.1$) – more details in Appendix A.

Figure 1 confirms the linear convergence of $J_l^N$ once the support is reached. However, the convergence might be unstable when the number of iteration grows, leading to exploding gradient, as illustrated in the second case. When this happens, using a small number of iterations or truncated back-propagation becomes necessary to prevent accumulating errors. It is also of interest to look at the proportion of unstable Jacobians (see Figure 2). We recover behaviors observed in the first and second case in Figure 1. 40% samples suffer from numerical instabilities in this example. This has a negative impact on the gradient estimation outside of the support.

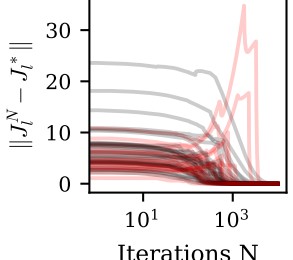

Figure 2: Average convergence of $J_l^N$ towards $J_l^*$ for 50 samples. In this example, 40% of the Jacobians are unstable (red curves).

We display the convergence behavior of the gradients estimated by AM and by DDL with different back-propagation depths (20, 50, full) for simulated data and images in Figure 3. We unroll FISTA instead of ISTA to make the convergence faster. We observed similar

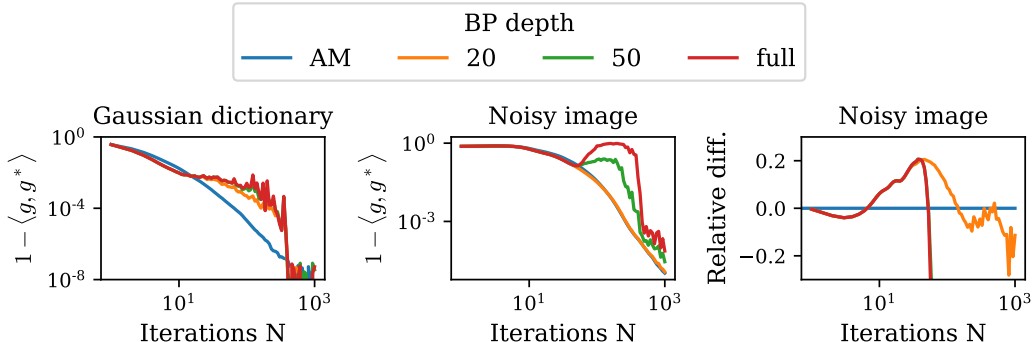

Figure 3: Gradient convergence in angle for 1000 synthetic samples (*left*) and patches from a noisy image (*center*). The image is normalized, decomposed into patches of dimension $10 \times 10$ and with additive Gaussian noise ($\sigma^2 = 0.1$). The dictionary for which the gradients are computed is composed of 128 patches from the image. (*right*) Relative difference between angles from DDL and AM. Convergence is faster with DDL in early iterations, and becomes unstable with too many steps.

behaviors for both algorithms in early iterations but using ISTA required too much memory to reach full convergence. As we optimize using a line search algorithm, we are mainly interested in the ability of the estimate to provide an adequate descent direction. Therefore, we display the convergence in angle defined as the cosine similarity $\langle \boldsymbol{g}, \boldsymbol{g}^* \rangle = \frac{Tr(\boldsymbol{g}^T \boldsymbol{g}^*)}{\|\boldsymbol{g}\|\|\boldsymbol{g}^*\|}$. The angle provides a good metric to assert that the two gradients are correlated and thus will lead to similar optimization paths. We also provide the convergence in norm in appendix. We compare $\boldsymbol{g}_N^1$ and $\boldsymbol{g}_N^2$ with the relative difference of their angles with $\boldsymbol{g}^*$, defined as $\frac{\langle \boldsymbol{g}_N^2, \boldsymbol{g}^* \rangle - \langle \boldsymbol{g}_N^1, \boldsymbol{g}^* \rangle}{1 - \langle \boldsymbol{g}_N^1, \boldsymbol{g}^* \rangle}$. When its value is positive, DDL provides the best descent direction. Generally, when the back-propagation goes too deep, the performance of $\boldsymbol{g}_N^2$ decreases compared to $\boldsymbol{g}_N^1$, and we observe large numerical instabilities. This behavior is coherent with the Jacobian convergence patterns studied in Proposition 2.5. Once on the support, $\boldsymbol{g}_N^2$ reaches back the performance of $\boldsymbol{g}_N^1$ as anticipated. In the case of a real image, unrolling beats AM by up to 20% in terms of gradient direction estimation when the number of iterations does not exceed 50, especially with small back-propagation depth. This highlights that the principal interest of unrolled algorithms is to use them with a small number of layers – i.e., a small number of iterations.

## 3 APPROXIMATE DICTIONARY LEARNING IN PRACTICE

This section introduces practical guidelines on Lasso-based approximate dictionary learning with unit norm constraint, and we provide empirical justifications for its ability to recover the dictionary. We also propose a strategy to scale DDL with a stochastic optimization method. We provide a full description of all our experiments in Appendix A. We optimize with projected gradient descent combined to a line search to compute high-quality steps sizes. The computations have been performed on a GPU NVIDIA Tesla V100-DGXS 32GB using `PyTorch` (Paszke et al., 2019).[1]

**Improvement of precision.** As stated before, a low number of iterations allows for efficient and stable computations, but this makes the sparse code less precise. One can learn the steps sizes of (F)ISTA to speed up convergence and compensate for imprecise representations, as proposed by Ablin et al. (2019) for LISTA. To avoid poor results due to large degrees of freedom in unsupervised learning, we propose a method in two steps to refine the initialization of the dictionary before relaxing the constraints on the steps sizes:

1. We learn the dictionary with fixed steps sizes equal to $\frac{1}{L}$ where $L = \|\boldsymbol{D}\|^2$, given by convergence conditions. Lipschitz constants or upper bounds are computed at each gradient step with norms, or the FFT for convolutions, outside the scope of the network graph.

2. Then, once convergence is reached, we jointly learn the step sizes and the dictionary. Both are still updated using gradient descent with line search to ensure stable optimization.

---

[1]Code is available at https://github.com/bmalezieux/unrolled_dl.

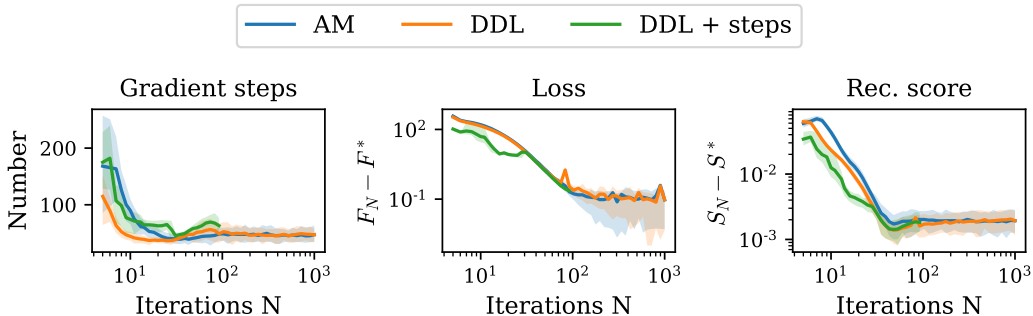

Figure 4: (*left*) Number of gradient steps performed by the line search before convergence, (*center*) distance to the optimal loss, and (*right*) distance to the optimal dictionary recovery score depending on the number of unrolled iterations. The data are generated as in Figure 1. We display the mean and the 10% and 90% quantiles over 50 random experiments. DDL needs less gradient steps to converge in early iterations, and unrolling obtains high recovery scores with only a few dozens of iterations.

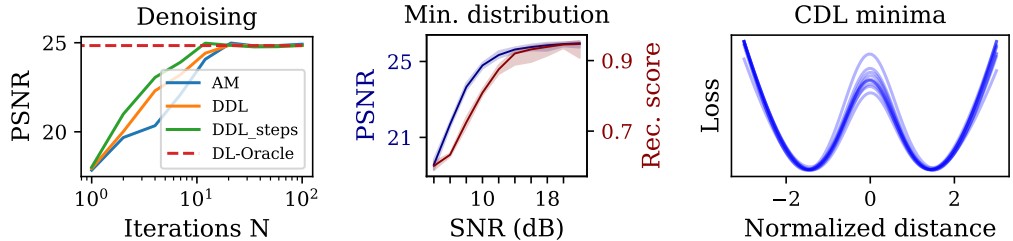

Figure 5: We consider a normalized image degraded by Gaussian noise. (*left*) PSNR depending on the number of unrolled iterations for $\sigma_{noise}^2 = 0.1$, i.e. PSNR = 10 dB. DL-Oracle stands for full AM dictionary learning ($10^3$ iterations of FISTA). There is no need to unroll too many iterations to obtain satisfying results. (*center*) PSNR and average recovery score between dictionaries depending on the SNR for 50 random initializations in CDL. (*right*) 10 loss landscapes in 1D for $\sigma_{noise}^2 = 0.1$. DDL is robust to random initialization when there is not too much noise.

The use of LISTA-like algorithms with no ground truth generally aims at improving the speed of sparse coding when high precision is not required. When it is the case, the final sparse codes can be computed separately with FISTA (Beck & Teboulle, 2009) or coordinate descent (Wu et al., 2008) to improve the quality of the representation.

## 3.1 OPTIMIZATION DYNAMICS IN APPROXIMATE DICTIONARY LEARNING

In this part, we study empirical properties of approximate dictionary learning related to global optimization dynamics to put our results on gradient estimation in a broader context.

**Unrolling v. AM.** In Figure 4, we show the number of gradient steps before reaching convergence, the behavior of the loss $F_N$, and the recovery score defined at the beginning of the section for synthetic data generated by a Gaussian dictionary. As a reminder, $S(C) = \max_{\sigma \in \mathfrak{S}_n} \frac{1}{n} \sum_{i=1}^n |C_{\sigma(i),i}|$ where $C$ is the correlation matrix between the columns of the true dictionary and the estimate. The number of iterations corresponds to $N$ in the estimate $z_N(D)$. First, DDL leads to fewer gradient steps than AM in the first iterations. This suggests that automatic differentiation better estimates the directions of the gradients for small depths. However, computing the gradient requires backpropagating through the algorithm, and DDL takes 1.5 times longer to perform one gradient step than AM on average for the same number of iterations $N$. When looking at the loss and the recovery score, we notice that the advantage of DDL for the minimization of $F_N$ is minor without learning the steps sizes, but there is an increase of performance concerning the recovery score. DDL better estimates the dictionary for small depths, inferior to 50. When unrolling more iterations, AM performs as well as DDL on the approximate problem and is faster.

**Approximate DL.** Figure 4 shows that high-quality dictionaries are obtained before the convergence of $F_N$, either with AM or DDL. 40 iterations are sufficient to reach a reasonable solution

concerning the recovery score, even though the loss is still very far from the optimum. This suggests that computing optimal sparse codes at each gradient step is unnecessary to recover the dictionary. Figure 5 illustrates that by showing the PSNR of a noisy image reconstruction depending on the number of iterations, compared to full AM dictionary learning with $10^3$ iterations. As for synthetic data, optimal performance is reached very fast. In this particular case, the model converges after 80 seconds with approximate DL unrolled for 20 iterations of FISTA compared to 600 seconds in the case of standard DL. Note that the speed rate highly depends on the value of $\lambda$. Higher values of $\lambda$ tend to make FISTA converge faster, and unrolling becomes unnecessary in this case. On the contrary, unrolling is more efficient than AM for lower values of $\lambda$.

**Loss landscape.** The ability of gradient descent to find adequate local minima strongly depends on the structure of the problem. To quantify this, we evaluate the variation of PSNR depending on the Signal to Noise Ratio (SNR) ($10 \log_{10} \left( \sigma^2 / \sigma_b^2 \right)$ where $\sigma_b^2$ is the variance of the noise) for 50 random initializations in the context of convolutional dictionary learning on a task of image denoising, with 20 unrolled iterations. Figure 5 shows that approximate CDL is robust to random initialization when the level of noise is not too high. In this case, all local minima are similar in terms of reconstruction quality. We provide a visualization of the loss landscape with the help of ideas presented in Li et al. (2018). The algorithm computes a minimum, and we chose two properly rescaled vectors to create a plan from this minimum. The 3D landscape is displayed on this plan in Figure B using the Python library K3D-Jupyter[2]. We also compare in Figure 5 (*right*) the shapes of local minima in 1D by computing the values of the loss along a line between two local minima. These visualizations confirm that dictionary learning locally behaves like a convex function with similar local minima.

## 3.2 STOCHASTIC DDL

In order to apply DDL in realistic settings, it is tempting to adapt Stochastic Gradient Descent (SGD), commonly used for neural networks. The major advantage is that the sparse coding is not performed on all data at each forward pass, leading to significant time and memory savings. The issue is that the choice of gradient steps is critical to the optimization process in dictionary learning, and SGD methods based on simple heuristics like rate decay are difficult to tune in this context. We propose to leverage a new optimization scheme introduced in Vaswani et al. (2019), which consists of performing a stochastic line search. The algorithm computes a good step size at each epoch, after which a heuristic decreases the maximal step. Figure 6 displays the recovery score function of the time for various mini-

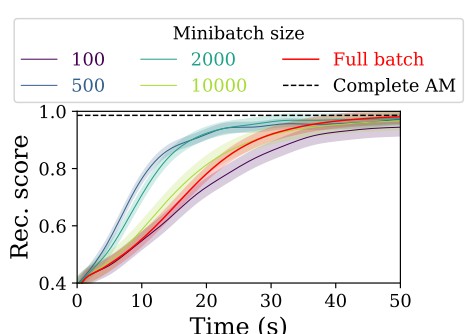

Figure 6: Recovery score vs. time for 10 random Gaussian matrices and $10^5$ samples. Initialization with random dictionaries. Intermediate batch sizes offer a good trade-off between speed and memory usage.

batch sizes on a problem with $10^5$ samples. The data were generated as in Figure 1 but with a larger dictionary ($50 \times 100$). The algorithm achieves good performance with small mini-batches and thus limited memory usage. We also compare this method with Online dictionary learning (Mairal et al., 2009) in Figure E. It shows that our method speeds up the dictionary recovery, especially for lower values of $\lambda$. This strategy can be adapted very easily for convolutional models by taking sub-windows of the full signal and performing a stochastic line search, as demonstrated in Section 4. See Tolooshams et al. (2020) for another unrolled stochastic CDL algorithm applied to medical data.

## 4 APPLICATION TO PATTERN LEARNING IN MEG SIGNALS

In magnetoencephalography (MEG), the measurements over the scalp consist of hundreds of simultaneous recordings, which provide information on the neural activity during a large period. Convolutional dictionary learning makes it possible to learn cognitive patterns corresponding to physiological activities (Dupré la Tour et al., 2018). As the electromagnetic waves propagate through the brain at the speed of light, every sensor measures the same waveform simultaneously but not at the same

---

[2]Package available at https://github.com/K3D-tools/K3D-jupyter.

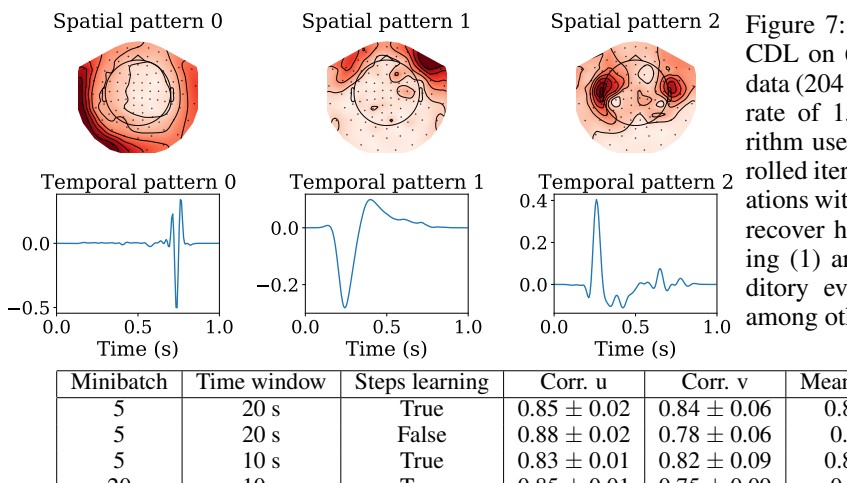

Figure 7: Stochastic Deep CDL on 6 minutes of MEG data (204 channels, sampling rate of 150Hz). The algorithm uses 40 atoms, 30 unrolled iterations and 100 iterations with batch size 20. We recover heartbeat (0), blinking (1) artifacts, and an auditory evoked response (2) among others.

| Minibatch | Time window | Steps learning | Corr. u | Corr. v | Mean corr. | Time |
|---|---|---|---|---|---|---|
| 5 | 20 s | True | $0.85 \pm 0.02$ | $0.84 \pm 0.06$ | 0.845 | 110 s |
| 5 | 20 s | False | $0.88 \pm 0.02$ | $0.78 \pm 0.06$ | 0.83 | 57 s |
| 5 | 10 s | True | $0.83 \pm 0.01$ | $0.82 \pm 0.09$ | 0.825 | 56 s |
| 20 | 10 s | True | $0.85 \pm 0.01$ | $0.75 \pm 0.09$ | 0.80 | 163 s |

Table 1: Stochastic Deep CDL on MEG data (as in Figure 7). We compare $u$ and $v$ to 12 important atoms output by `alphacsc` (correlation averaged on 5 runs), depending on several hyperparameters, with 30 layers, 10 epochs and 10 iterations per epochs. $\lambda_{\text{rescaled}} = 0.3\lambda_{\max}$, $\lambda_{\max} = \left\| D^T y \right\|_\infty$. The best setups achieve $80\% - 90\%$ average correlation with `alphacsc` in around 100 sec. compared to around 1400 sec. Our method is also faster than convolutional K-SVD (Yellin et al., 2017).

intensity. The authors propose to rely on multivariate convolutional sparse coding (CSC) with rank-1 constraint to leverage this physical property and learn prototypical patterns. In this case, space and time patterns are disjoint in each atom: $\boldsymbol{D}_k = \boldsymbol{u}_k \boldsymbol{v}_k^T$ where $\boldsymbol{u}$ gathers the spatial activations on each channel and $\boldsymbol{v}$ corresponds to the temporal pattern. This leads to the model

$$\min_{\boldsymbol{z}_k \in \mathbb{R}^T, \boldsymbol{u}_k \in \mathbb{R}^S, \boldsymbol{v}_k \in \mathbb{R}^t} \frac{1}{2} \left\| \sum_{k=1}^n (\boldsymbol{u}_k \boldsymbol{v}_k^\top) * \boldsymbol{z}_k - \boldsymbol{y} \right\|_2^2 + \lambda \sum_{k=1}^n \|\boldsymbol{z}_k\|_1 \quad , \tag{6}$$

where $n$ is the number of atoms, $T$ is the total recording time, $t$ is the kernel size, and $S$ is the number of sensors. We propose to learn $\boldsymbol{u}$ and $\boldsymbol{v}$ with Stochastic Deep CDL unrolled for a few iterations to speed up the computations of the atoms. Figure 7 reproduces the multivariate CSC experiments of `alphacsc`[3] (Dupré la Tour et al., 2018) on the dataset sample of MNE (Gramfort et al., 2013) – 6 minutes of recordings with 204 channels sampled at 150Hz with visual and audio stimuli.

The algorithm recovers the main waveforms and spatial patterns with approximate sparse codes and without performing the sparse coding on the whole data set at each gradient iteration, which leads to a significant gain of time. We are able to distinguish several meaningful patterns as heartbeat and blinking artifacts or auditive evoked response. As this problem is unsupervised, it is difficult to provide robust quantitative quality measurements. Therefore, we compare our patterns to 12 important patterns recovered by alpahcsc in terms of correlation in Table 1. Good setups achieve between 80% and 90% average correlation ten times faster.

## 5 CONCLUSION

Dictionary learning is an efficient technique to learn patterns in a signal but is challenging to apply to large real-world problems. This work showed that approximate dictionary learning, which consists in replacing the optimal solution of the Lasso with a time-efficient approximation, offers a valuable trade-off between computational cost and quality of the solution compared to complete Alternating Minimization. This method, combined with a well-suited stochastic gradient descent algorithm, scales up to large data sets, as demonstrated on a MEG pattern learning problem. This work provided a theoretical study of the asymptotic behavior of unrolling in approximate dictionary learning. In particular, we showed that numerical instabilities make DDL usage inefficient when too many iterations are unrolled. However, the super-efficiency of DDL in the first iterations remains unexplained, and our first findings would benefit from theoretical support.

---

[3]Package and experiments available at `https://alphacsc.github.io`

ETHICS STATEMENT

The MEG data conform to ethic guidelines (no individual names, collected under individual's consent, . . . ).

REPRODUCIBILITY STATEMENT

Code is available at https://github.com/bmalezieux/unrolled_dl. We provide a full description of all our experiments in Appendix A, and the proofs of our theoretical results in Appendix C.

ACKNOWLEDGMENTS

This work was supported by grants from Digiteo France.

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

## A   FULL DESCRIPTION OF THE EXPERIMENTS

This section provides complementary information on the experiments presented in the paper.

### A.1   CONVERGENCE OF THE JACOBIANS - FIGURE 1 AND FIGURE 2

We generate a normalized random Gaussian dictionary $D$ of dimension $30 \times 50$, and sparse codes $z$ from a Bernoulli Gaussian distribution of sparsity 0.3 and $\sigma^2 = 1$. The signal to process is $y = Dz + b$ where $b$ is an additive Gaussian noise with $\sigma^2_{noise} = 0.1$. The Jacobians are computed for a random perturbation $D + b_D$ of $D$ where $b_D$ is a Gaussian noise of scale $0.5\sigma^2_D$. $J_l^N$ corresponds to the approximate Jacobian with N iterations of ISTA with $\lambda = 0.1$. $J_l^*$ corresponds the true Jacobian computed with sparse codes obtained after $10^4$ iterations of ISTA with $\lambda = 0.1$.

In Figure 2, the norm $\left\| J_l^N - J_l^* \right\|$ is computed for 50 samples.

### A.2   CONVERGENCE OF THE GRADIENT ESTIMATES - FIGURE 3

**Synthetic data.**   We generate a normalized random Gaussian dictionary $D$ of dimension $30 \times 50$, and 1000 sparse codes $z$ from a Bernoulli Gaussian distribution of sparsity 0.3 and $\sigma^2 = 1$. The signal to process is $y = Dz + b$ where $b$ is an additive Gaussian noise with $\sigma^2_{noise} = 0.1$. The gradients are computed for a random perturbation $D + b_D$ of $D$ where $b_D$ is a Gaussian noise of scale $0.5\sigma^2_D$.

**Noisy image.**   A $128 \times 128$ black-and-white image is degraded by a Gaussian noise with $\sigma^2_{noise} = 0.1$ and normalized. We processed 1000 patches of dimension $10 \times 10$ from the image, and we computed the gradients for a dictionary composed of 128 random patches.

$g_N$ corresponds to the gradient for N iterations of FISTA with $\lambda = 0.1$. $g^*$ corresponds to the true gradient computed with a sparse code obtained after $10^4$ iterations of FISTA.

### A.3   OPTIMIZATION DYNAMICS ON SYNTHETIC DATA - FIGURE 4

We generate a normalized random Gaussian dictionary $D$ of dimension $30 \times 50$, and sparse codes $z$ from a Bernoulli Gaussian distribution of sparsity 0.3 and $\sigma^2 = 1$. The signal to process is $y = Dz + b$ where $b$ is an additive Gaussian noise with $\sigma^2_{noise} = 0.1$. The initial dictionary

is taken as a random perturbation $\boldsymbol{D} + \boldsymbol{b_D}$ of $\boldsymbol{D}$ where $\boldsymbol{b_D}$ is a Gaussian noise of scale $0.5\sigma_D^2$. $N$ corresponds to the number of unrolled iterations of FISTA. $F^*$ is the value of the loss for $10^3$ iterations minus $10^{-3}$. $S^*$ is the score obtained after $10^3$ iterations plus $10^{-3}$. The optimization is done with $\lambda = 0.1$. We compare the number of gradient steps (left), the loss values (center), and the recovery scores (right) for 50 different dictionaries. DDL with steps sizes learning is evaluated on 100 iterations only due to memory and optimization time issues.

## A.4 Optimization dynamics and loss landscapes on images - Figure 5

A $128 \times 128$ black-and-white image is degraded by a Gaussian noise and normalized.

**Left.** In this experiment, $\sigma_{\text{noise}}^2 = 0.1$. We learn a dictionary composed of 128 atoms on $10 \times 10$ patches with FISTA and $\lambda = 0.1$ in all cases. The PSNR is obtained with sparse codes output by the network. The results are compared to the truth with the Peak Signal to Noise Ratio. Dictionary learning denoising with 1000 iterations of FISTA is taken as a baseline.

**Center.** We learn 50 dictionaries from 50 random initializations in convolutional dictionary learning with 50 kernels of size $8 \times 8$ with 20 unrolled iterations of FISTA and $\lambda = 0.1$. The PSNR is obtained with sparse codes output by the network. We compare the average, minimal and maximal PSNR, and recovery scores with all other dictionaries to study the robustness to random initialization depending on the level of noise (SNR).

**Right.** In this experiment, $\sigma_{\text{noise}}^2 = 0.1$. We learn 2 dictionaries from 2 random initializations in convolutional dictionary learning with 50 kernels of size $8 \times 8$ with 20 unrolled iterations of FISTA and $\lambda = 0.1$. We display the loss values on the line between these two dictionaries. The experiment is repeated on 10 different random initializations.

## A.5 Stochastic DDL on synthetic data - Figure 6

We generate a normalized random Gaussian dictionary $D$ of dimension $50 \times 100$, and $10^5$ sparse codes $z$ from a Bernoulli Gaussian distribution of sparsity 0.3 and $\sigma^2 = 1$. The signal to process is $y = Dz + b$ where $b$ is an additive Gaussian noise with $\sigma_{noise}^2 = 0.1$. The initial dictionary is taken as a random gaussian dictionary. We compare stochastic and full-batch line search projected gradient descent with 30 unrolled iterations of FISTA and $\lambda = 0.1$, without steps sizes learning. Stochastic DDL is run for 10 epochs with a maximum of 100 iterations for each epoch.

## A.6 Pattern learning in MEG - Figure 7

Stochastic Deep CDL on 6 minutes of recordings of MEG data with 204 channels and a sampling rate of 150Hz. We remove the powerline artifacts and high-pass filter the signal to remove the drift which can impact the CSC technique. The signal is also resampled to 150 Hz to reduce the computational burden. This preprocessing procedure is presented in `alphacsc`, and available in the code in the supplementary materials. The algorithm learns 40 atoms of 1 second on mini batches of 10 seconds, with 30 unrolled iterations of FISTA, $\lambda_{\text{scaled}} = 0.3$, and 10 epochs with 10 iterations per epoch. The number of mini-batches per iteration is 20, with possible overlap.

## B Extra figures and experimental results

**LISTA - Figure A.** Illustration of LISTA for Dictionary Learning with initialization $\boldsymbol{Z}_0 = 0$ for $N = 3$. $\boldsymbol{W}_{\boldsymbol{D}}^1 = \frac{1}{L}(\boldsymbol{D})^\top$, $\boldsymbol{W}_{\boldsymbol{D}}^2 = (I - \frac{1}{L}(\boldsymbol{D})^\top \boldsymbol{D})$, where $L = \|\boldsymbol{D}\|^2$. The result $\boldsymbol{Z}_N(\boldsymbol{D})$ output by the network is an approximation of the solution of the LASSO.

**Loss landscape in 2D - Figure B.** We provide a visualization of the loss landscape with the help of ideas presented in Li et al. (2018). The algorithm computes a minimum, and we chose two properly rescaled vectors to create a plan from this minimum. The 3D landscape is displayed on this plan in the appendix using the Python library K3D-Jupyter. This visualization and the visualization in 1D confirm that (approximate) dictionary learning locally behaves like a convex function with smooth local minima.

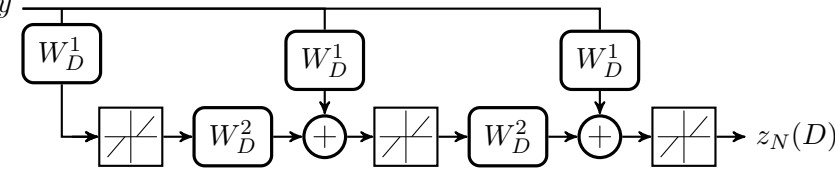

Figure A: LISTA

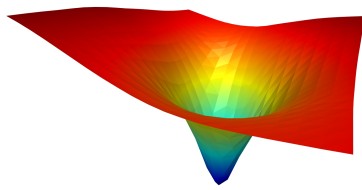

Figure B: Loss landscape in approximate CDL

**Gradient convergence in norm - Figure C.** Gradient estimates convergence in norm for synthetic data (*left*) and patches from a noisy image (*right*). The setup is similar to Figure 3. Both gradient estimates converge smoothly in early iterations. When the back-propagation goes too deep, the performance of $g_N^2$ decreases compared to $g_N^1$, and we observe large numerical instabilities. This behavior is coherent with the Jacobian convergence patterns studied in Proposition 2.5. Once on the support, $g_N^2$ reaches back the performance of $g_N^1$.

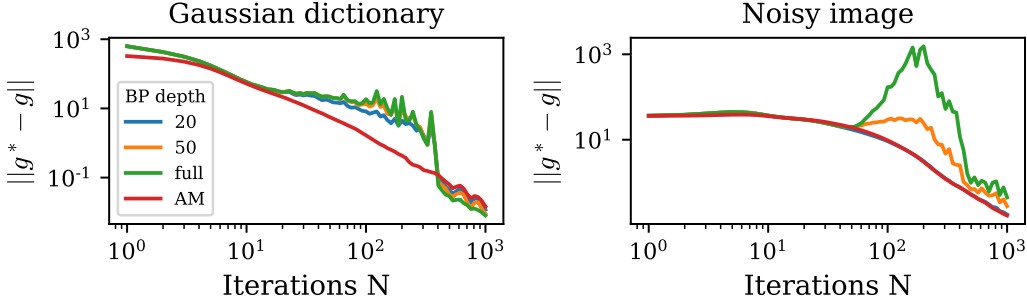

Figure C: Gradient estimates convergence in norm for synthetic data (*left*) and patches from a noisy image (*right*). Both gradient estimates converge smoothly in early iterations, after what DDL gradient becomes unstable. The behavior returns to normal once the algorithm reaches the support.

**Computation time to reach 0.95 recovery score - Figure D.** The setup is similar to Figure 6. A random Gaussian dictionary of size $50 \times 100$ generates the data from $10^5$ sparse codes with sparsity 0.3. The approximate sparse coding is solved with $\lambda = 0.1$ and 30 unrolled iterations of FISTA. The algorithm achieves good performances with small mini-batches and thus limited memory usage. Stochastic DDL can process large amounts of data and recovers good quality dictionaries faster than full batch DDL.

**Sto DDL vs. Online DL - Figure E.** We compare the time Online DL from `spams`[4] (Mairal et al., 2009) and Stochastic DDL need to reach a recovery score of 0.95 with a batch size of 2000. Online DL is run with 10 threads. We repeat the experiment 10 times for different values of $\lambda$ from

---

[4]package available at http://thoth.inrialpes.fr/people/mairal/spams/

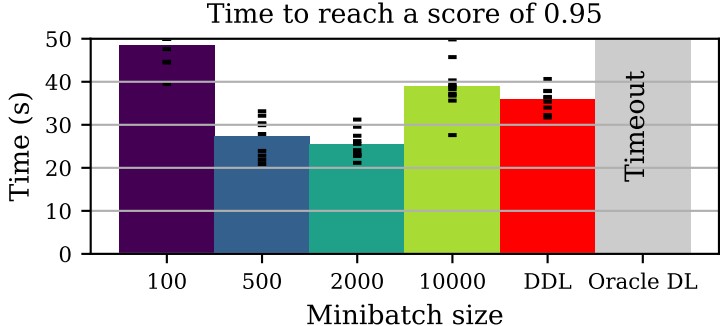

Figure D: Time to reach a recovery score of 0.95. Intermediate batch sizes offer a good trade-off between speed and memory usage compared to full-batch DDL.

0.1 to 1.0. The setup is similar to Figure D, and we initialize both methods randomly. Stochastic DDL is more efficient for smaller values of $\lambda$, due to the fact that sparse coding is slower in this case. For higher values of $\lambda$, both methods are equivalent. Another advantage of Stochastic DDL is its modularity. It works on various kinds of dictionary parameterization thanks to automatic differentiation, as illustrated on 1-rank multivariate convolutional dictionary learning in Figure 7.

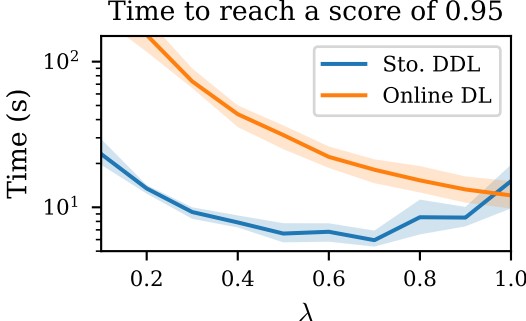

Figure E: Comparison between Online DL and Stochastic DDL. Stochastic DDL is more efficient for smaller values of $\lambda$, due to the fact that sparse coding is slower in this case.

## C    Proofs of theoretical results

This section gives the proofs for the various theoretical results in the paper.

### C.1    Proof of Proposition 2.1.

**Proposition 2.1** *Let* $\boldsymbol{D}^* = \arg\min_{\boldsymbol{D}\in\mathcal{C}} G(\boldsymbol{D})$ *and* $\boldsymbol{D}_N^* = \arg\min_{\boldsymbol{D}\in\mathcal{C}} G_N(\boldsymbol{D})$, *where* $N$ *is the number of unrolled iterations. We denote by* $K(\boldsymbol{D}^*)$ *a constant depending on* $\boldsymbol{D}^*$, *and by* $C(N)$ *the convergence speed of the algorithm, which approximates the inner problem solution. We have*

$$G_N(\boldsymbol{D}_N^*) - G(\boldsymbol{D}^*) \leq K(\boldsymbol{D}^*)C(N) \ .$$

Let $G(\boldsymbol{D}) \triangleq F(\boldsymbol{Z}^*(\boldsymbol{D}), \boldsymbol{D})$ and $G_N(\boldsymbol{D}) \triangleq F(\boldsymbol{Z}_N(\boldsymbol{D}), \boldsymbol{D})$ where $\boldsymbol{Z}^*(\boldsymbol{D}) = \arg\min_{\boldsymbol{Z}\in\mathbb{R}^{n\times T}} F(\boldsymbol{Z}, \boldsymbol{D})$ and $\boldsymbol{Z}_N(\boldsymbol{D}) = FISTA(\boldsymbol{D}, N)$. Let $\boldsymbol{D}^* = \arg\min_{\boldsymbol{D}\in\mathcal{C}} G(\boldsymbol{D})$ and $\boldsymbol{D}_N^* = \arg\min_{\boldsymbol{D}\in\mathcal{C}} G_N(\boldsymbol{D})$. We have

$$G_N(\boldsymbol{D}_N^*) - G(\boldsymbol{D}^*) = G_N(\boldsymbol{D}_N^*) - G_N(\boldsymbol{D}^*) + G_N(\boldsymbol{D}^*) - G(\boldsymbol{D}^*) \tag{7}$$
$$= F(\boldsymbol{Z}_N(\boldsymbol{D}_N), \boldsymbol{D}_N) - F(\boldsymbol{Z}_N(\boldsymbol{D}^*), \boldsymbol{D}^*) \tag{8}$$
$$+ F(\boldsymbol{Z}_N(\boldsymbol{D}^*), \boldsymbol{D}^*) - F(\boldsymbol{Z}(\boldsymbol{D}^*), \boldsymbol{D}^*) \tag{9}$$

By definition of $\boldsymbol{D}_N^*$

$$F(\boldsymbol{Z}_N(\boldsymbol{D}_N^*), \boldsymbol{D}_N^*) - F(\boldsymbol{Z}_N(\boldsymbol{D}^*), \boldsymbol{D}^*) \leq 0 \tag{10}$$

The convergence rate of FISTA in function value for a fixed dictionary $\boldsymbol{D}$ is

$$F(\boldsymbol{Z}_N(\boldsymbol{D}), \boldsymbol{D}) - F(\boldsymbol{Z}_N(\boldsymbol{D}), \boldsymbol{D}) \leq \frac{K(\boldsymbol{D})}{N^2} \tag{11}$$

Therefore

$$F(\boldsymbol{Z}_N(\boldsymbol{D}^*), \boldsymbol{D}^*) - F(\boldsymbol{Z}(\boldsymbol{D}^*), \boldsymbol{D}^*) \leq \frac{K(\boldsymbol{D}^*)}{N^2} \tag{12}$$

Hence

$$G_N(\boldsymbol{D}_N^*) - G(\boldsymbol{D}^*) \leq \frac{K(\boldsymbol{D}^*)}{N^2} \tag{13}$$

## C.2 PROOF OF PROPOSITION 2.2

**Proposition 2.2** *Let $\boldsymbol{D} \in \mathbb{R}^{m \times n}$. Then, there exists a constant $L_1 > 0$ such that for every number of iterations $N$*

$$\left\| \boldsymbol{g}_N^1 - \boldsymbol{g}^* \right\| \leq L_1 \left\| \boldsymbol{z}_N(\boldsymbol{D}) - \boldsymbol{z}^*(\boldsymbol{D}) \right\| \ .$$

We have

$$F(\boldsymbol{z}, \boldsymbol{D}) = \frac{1}{2} \left\| \boldsymbol{D}\boldsymbol{z} - \boldsymbol{y} \right\|_2^2 + \lambda \left\| \boldsymbol{z} \right\|_1 \tag{14}$$

$$\nabla_2 F(\boldsymbol{z}, \boldsymbol{D}) = (\boldsymbol{D}\boldsymbol{z} - \boldsymbol{y})\boldsymbol{z}^\top \tag{15}$$

$\boldsymbol{z}_0(\boldsymbol{D}) = 0$ and the iterates $(\boldsymbol{z}_N(\boldsymbol{D}))_{N \in \mathbb{N}}$ converge towards $\boldsymbol{z}^*(\boldsymbol{D})$. Hence, they are contained in a closed ball around $\boldsymbol{z}^*(\boldsymbol{D})$. As $\nabla_2 F(\cdot, \boldsymbol{D})$ is continuously differentiable, it is locally Lipschitz on this closed ball, and there exists a constant $L_1(\boldsymbol{D})$ depending on $\boldsymbol{D}$ such that

$$\left\| \boldsymbol{g}_N^1 - \boldsymbol{g}^* \right\| = \left\| \nabla_2 F(\boldsymbol{z}_N(\boldsymbol{D}), \boldsymbol{D}) - \nabla_2 F(\boldsymbol{z}^*(\boldsymbol{D}), \boldsymbol{D}) \right\| \tag{16}$$

$$\leq L_1(\boldsymbol{D}) \left\| \boldsymbol{z}_N(\boldsymbol{D}) - \boldsymbol{z}^*(\boldsymbol{D}) \right\| \tag{17}$$

## C.3 PROOF OF PROPOSITION 2.3.

**Proposition 2.3** *Let $\boldsymbol{D} \in \mathbb{R}^{m \times n}$. Let $S^*$ be the support of $\boldsymbol{z}^*(\boldsymbol{D})$, $S_N$ be the support of $\boldsymbol{z}_N$ and $\widetilde{S}_N = S_N \cup S^*$. Let $f(\boldsymbol{z}, \boldsymbol{D}) = \frac{1}{2} \left\| \boldsymbol{D}\boldsymbol{z} - \boldsymbol{y} \right\|_2^2$ be the data-fitting term in $F$. Let $R(\mathbf{J}, \widetilde{S}) = \mathbf{J}^+ \left( \nabla_{1,1}^2 f(\boldsymbol{z}^*, \boldsymbol{D}) \odot \mathbb{1}_{\widetilde{S}} \right) + \nabla_{2,1}^2 f(\boldsymbol{z}^*, \boldsymbol{D}) \odot \mathbb{1}_{\widetilde{S}}$. Then there exists a constant $L_2 > 0$ and a subsequence of (F)ISTA iterates $\boldsymbol{z}_{\phi(N)}$ such that for all $N \in \mathbb{N}$:*

$$\exists \, \boldsymbol{g}_{\phi(N)}^2 \in \nabla_2 f(\boldsymbol{z}_{\phi(N)}, \boldsymbol{D}) + \mathbf{J}_{\phi(N)}^+ \left( \nabla_1 f(\boldsymbol{z}_{\phi(N)}, \boldsymbol{D}) + \lambda \partial_{\|\cdot\|_1}(\boldsymbol{z}_{\phi(N)}) \right) \, s.t. :$$

$$\left\| \boldsymbol{g}_{\phi(N)}^2 - \boldsymbol{g}^* \right\| \leq \left\| R(\mathbf{J}_{\phi(N)}, \widetilde{S}_{\phi(N)}) \right\| \left\| \boldsymbol{z}_{\phi(N)} - \boldsymbol{z}^* \right\| + \frac{L_2}{2} \left\| \boldsymbol{z}_{\phi(N)} - \boldsymbol{z}^* \right\|^2 \ .$$

*This sub-sequence $\boldsymbol{z}_{\phi(N)}$ corresponds to iterates on the support of $\boldsymbol{z}^*$.*

We have

$$\boldsymbol{g}_N^2(\boldsymbol{D}) \in \nabla_2 f(\boldsymbol{z}_N(\boldsymbol{D}), \boldsymbol{D}) + \mathbf{J}_N^+ \left( \nabla_1 f(\boldsymbol{z}_N(\boldsymbol{D}), \boldsymbol{D}) + \lambda \partial_{\|\cdot\|_1}(\boldsymbol{z}_N) \right) \tag{18}$$

We adapt equation (6) in Ablin et al. (2020)

$$\boldsymbol{g}_N^2 = \boldsymbol{g}^* + R(\mathbf{J}_N, \widetilde{S_N})(\boldsymbol{z}_N - \boldsymbol{z}^*) + R_N^{\boldsymbol{D}, \boldsymbol{z}} + \mathbf{J}_N^+ R_N^{\boldsymbol{z}, \boldsymbol{z}} \tag{19}$$

where

$$R(\mathbf{J}, \widetilde{S}) = \mathbf{J}^+ \left( \nabla_{1,1}^2 f(\boldsymbol{z}^*, \boldsymbol{D}) \odot \mathbb{1}_{\widetilde{S}} \right) + \nabla_{2,1}^2 f(\boldsymbol{z}^*, \boldsymbol{D}) \odot \mathbb{1}_{\widetilde{S}} \tag{20}$$

$$R_N^{\boldsymbol{D}, \boldsymbol{z}} = \nabla_2 f(\boldsymbol{z}_N, \boldsymbol{D}) - \nabla_2 f(\boldsymbol{z}^*, \boldsymbol{D}) - \nabla_{2,1}^2 f(\boldsymbol{z}^*, \boldsymbol{D})(\boldsymbol{z}_N - \boldsymbol{z}^*) \tag{21}$$

$$R_N^{\boldsymbol{z}, \boldsymbol{z}} \in \nabla_1 f(\boldsymbol{z}_N, \boldsymbol{D}) + \lambda \partial_{\|\cdot\|_1}(\boldsymbol{z}_N) - \nabla_{1,1}^2 f(\boldsymbol{z}^*, \boldsymbol{D})(\boldsymbol{z}_N - \boldsymbol{z}^*) \tag{22}$$

As $z_N$ and $z^*$ are on $\widetilde{S_N}$

$$\nabla_{2,1}^2 f(z^*, D)(z_N - z^*) = \left( \nabla_{2,1}^2 f(z^*, D) \odot \mathbb{1}_{\widetilde{S_N}} \right)(z_N - z^*) \tag{23}$$

$$\mathbf{J}^+ \left( \nabla_{1,1}^2 f(z^*, D)(z_N - z^*) \right) = \mathbf{J}^+ \left( \nabla_{1,1}^2 f(z^*, D) \odot \mathbb{1}_{\widetilde{S_N}}(z_N - z^*) \right) \tag{24}$$

As stated in Proposition 2.2, $\nabla_2 f(\cdot, D)$ is locally Lipschitz, and $R_N^{D,z}$ is the Taylor rest of $\nabla_2 f(\cdot, D)$. Therefore, there exists a constant $L_{D,z}$ such that

$$\forall N \in \mathbb{N}, \left\| R_N^{D,z} \right\| \leq \frac{L_{D,z}}{2} \left\| z_N(D) - z^*(D) \right\|^2 \tag{25}$$

We know that $0 \in \nabla_1 f(z^*, D) + \lambda \partial_{\|\cdot\|_1}(z^*)$. In other words, $\exists u^* \in \lambda \partial_{\|\cdot\|_1}(z^*)$ s.t. $\nabla_1 f(z^*, D) + u^* = 0$. Therefore we have:

$$R_N^{z,z} \in \nabla_1 f(z_N, D) - \nabla_1 f(z^*, D) - \nabla_{1,1}^2 f(z^*, x)(z_N - z^*) + \lambda \partial \|z_N\|_1 - u^* \tag{26}$$

Let $L_{z,z}$ be the Lipschitz constant of $\nabla_1 f(\cdot, D)$. (F)ISTA outputs a sequence such that there exists a sub-sequence $(z_{\phi(N)})_{N \in \mathbb{N}}$ which has the same support as $z^*$. For this sub-sequence, $u^* \in \lambda \partial_{\|\cdot\|_1}(z_{\phi(N)})$. Therefore, there exists $R_{\phi(N)}^{z,z}$ such that

1. $R_{\phi(N)}^{z,z} \in \nabla_1 f(z_{\phi(N)}, D) + \lambda \partial_{\|\cdot\|_1}(z_{\phi(N)}) - \nabla_{1,1}^2 f(z^*, x)(z_{\phi(N)} - z^*)$

2. $\left\| R_{\phi(N)}^{z,z} \right\| \leq \frac{L_{z,z}}{2} \left\| z_{\phi(N)} - z^* \right\|^2$

For this sub-sequence, we can adapt Proposition 2 from Ablin et al. (2020). Let $L_2 = L_{D,z} + L_{z,z}$, we have

$$\exists g_{\phi(N)}^2 \in \nabla_2 f(z_{\phi(N)}, D) + \mathbf{J}_{\phi(N)} \left( \nabla_1 f(z_{\phi(N)}, D) + \lambda \partial \left\| z_{\phi(N)} \right\|_1 \right), \text{ s.t. :} \tag{27}$$

$$\left\| g_{\phi(N)}^2 - g^* \right\| \leq \left\| R(\mathbf{J}_{\phi(N)}, \widetilde{S_{\phi(N)}}) \right\| \left\| z_{\phi(N)} - z^* \right\| + \frac{L_2}{2} \left\| z_{\phi(N)} - z^* \right\|^2 \tag{28}$$

## C.4   Proof of Theorem 2.4.

**Theorem 2.4** *At iteration $N + 1$ of ISTA, the weak Jacobian of $z_{N+1}$ relatively to $D_l$, where $D_l$ is the $l$-th row of $D$, is given by induction:*

$$\frac{\partial(z_{N+1})}{\partial D_l} = \mathbb{1}_{|z_{N+1}|>0} \odot \left( \frac{\partial(z_N)}{\partial D_l} - \frac{1}{L} \left( D_l z_N^\top + (D_l^\top z_N - y_l) I_n + D^\top D \frac{\partial(z_N)}{\partial D_l} \right) \right) \ .$$

*$\frac{\partial(z_N)}{\partial D_l}$ will be denoted by $J_l^N$. It converges towards the weak Jacobian $J_l^*$ of $z^*$ relatively to $D_l$, whose values are*

$$J_l^* {}_{S^*} = -(D_{:,S^*}^\top D_{:,S^*})^{-1} (D_l z^{*\top} + (D_l^\top z^* - y_l) I_n)_{S^*} \ ,$$

*on the support $S^*$ of $z^*$, and 0 elsewhere. Moreover, $R(\mathbf{J}^*, S^*) = 0$.*

We start by recalling a Lemma from Deledalle et al. (2014).

**Lemma C.1** *The soft-thresholding $ST_\mu$ defined by $ST_\mu(z) = sgn(z) \odot (|z| - \mu)_+$ is weakly differentiable with weak derivative $\frac{dST_\mu(z)}{dz} = \mathbb{1}_{|z|>\mu}$.*

Coordinate-wise, ISTA corresponds to the following equality:

$$z_{N+1} = ST_\mu((I - \frac{1}{L} D^\top D) z_N + \frac{1}{L} D^\top y) \tag{29}$$

$$(z_{N+1})_i = ST_\mu((z_N)_i - \frac{1}{L} \sum_{p=1}^m (\sum_{j=1}^n D_{ji} D_{jp})(z_N)_p + \frac{1}{L} \sum_{j=1}^n D_{ji} y_j) \tag{30}$$

The Jacobian is computed coordinate wise with the chain rule:

$$\frac{\partial(z_{N+1})_i}{\partial D_{lk}} = \mathbb{1}_{|(z_{N+1})_i|>0} \cdot \left( \frac{\partial(z_N)_i}{\partial D_{lk}} - \frac{1}{L}\frac{\partial}{\partial D_{lk}}(\sum_{p=1}^{m}(\sum_{j=1}^{n} D_{ji}D_{jp})(z_N)_p) + \frac{1}{L}\frac{\partial}{\partial D_{lk}}\sum_{j=1}^{n} D_{ji}y_j) \right) \tag{31}$$

Last term:

$$\frac{\partial}{\partial D_{lk}}\sum_{j=1}^{n} D_{ji}y_j = \delta_{ik}y_l \tag{32}$$

Second term:

$$\frac{\partial}{\partial D_{lk}}\sum_{p=1}^{m}\sum_{j=1}^{n} D_{ji}D_{jp}(z_N)_p = \sum_{p=1}^{m}\sum_{j=1}^{n} D_{ji}D_{jp}\frac{\partial(z_N)_p}{\partial D_{lk}} + \sum_{p=1}^{m}\sum_{j=1}^{n} \frac{\partial D_{ji}D_{jp}}{\partial D_{lk}}(z_N)_p \tag{33}$$

$$\frac{\partial D_{ji}D_{jp}}{\partial D_{lk}} = \begin{cases} 2D_{lk} & \text{if } j=l \text{ and } i=p=k \\ D_{lp} & \text{if } j=l \text{ and } i=k \text{ and } p \neq k \\ D_{li} & \text{if } j=l \text{ and } i \neq k \text{ and } p=k \\ 0 & \text{else} \end{cases} \tag{34}$$

Therefore:

$$\sum_{p=1}^{m}\sum_{j=1}^{n} \frac{\partial D_{ji}D_{jp}}{\partial D_{lk}}(z_N)_p = \sum_{p=1}^{m}(2D_{lk}\delta_{ip}\delta_{ik} + D_{li}\delta_{pk}\mathbb{1}_{i\neq k} + D_{lp}\delta_{ik}\mathbb{1}_{k\neq p})(z_N)_p \tag{35}$$

$$= 2D_{lk}(z_N)_k\delta_{ik} + D_{li}(z_N)_k\mathbb{1}_{i\neq k} + \sum_{\substack{p=1 \\ p\neq k}}^{m} D_{lp}(z_N)_p\delta_{ik} \tag{36}$$

$$= D_{li}(z_N)_k + \delta_{ik}\sum_{p=1}^{m} D_{l_p}(z_N)_p \tag{37}$$

Hence:

$$\frac{\partial(z_{N+1})_i}{\partial D_{lk}} = \mathbb{1}_{|(z_{N+1})_i|>0} \cdot \left( \frac{\partial(z_N)_i}{\partial D_{lk}} - \frac{1}{L}(D_{li}(z_N)_k + \right. \tag{38}$$

$$\left. \delta_{ik}(\sum_{p=1}^{m} D_{lp}(z_N)_p) + \sum_{p=1}^{m}\sum_{j=1}^{n} \frac{\partial(z_N)_p}{\partial D_{lk}}D_{ji}D_{jp} - \delta_{ik}y_l) \right)$$

This leads to the following vector formulation:

$$\frac{\partial(\boldsymbol{z}_{N+1})}{\partial D_l} = \mathbb{1}_{|\boldsymbol{z}_{N+1}|>0} \odot \left( \frac{\partial(\boldsymbol{z}_N)}{\partial D_l} - \frac{1}{L}\left( D_l\boldsymbol{z}_N^\top + (D_l^\top \boldsymbol{z}_N - y_l)\boldsymbol{I}_m + \boldsymbol{D}^\top \boldsymbol{D}\frac{\partial(\boldsymbol{z}_N)}{\partial D_l} \right) \right) \tag{39}$$

On the support of $\boldsymbol{z}^*$, denoted by $S^*$, this quantity converges towards the fixed point:

$$J_l^* = -(D_{:,S^*}^\top D_{:,S^*})^{-1}(D_l z^{*\top} + (D_l^\top \boldsymbol{z}^* - y_l)\boldsymbol{I}_m)_{S^*} \tag{40}$$

Elsewhere, $J_l^*$ is equal to 0. To prove that $R(\mathbf{J}^*, S^*) = 0$, we use the expression given by equation 39

$$\mathbf{J}^* = \mathbb{1}_{S^*} \odot \left( \mathbf{J}^* - \frac{1}{L}\left( \nabla_{2,1}^2 f(\boldsymbol{z}^*, \boldsymbol{D}_l)^\top + \nabla_{1,1}^2 f(\boldsymbol{z}^*, \boldsymbol{D})^\top \mathbf{J}^* \right) \right) \tag{41}$$

$$\mathbf{J}^* - \mathbb{1}_{S^*} \odot \mathbf{J}^* = \frac{1}{L}\mathbb{1}_{S^*} \odot \nabla_{2,1}^2 f(\boldsymbol{z}^*, \boldsymbol{D}_l)^\top + \mathbb{1}_{S^*} \odot \nabla_{1,1}^2 f(\boldsymbol{z}^*, \boldsymbol{D})^\top \mathbf{J}^* \tag{42}$$

$$0 = \mathbf{J}^{*+}\left( \nabla_{1,1}^2 f(\boldsymbol{z}^*, \boldsymbol{D}) \odot \mathbb{1}_{S^*} \right) + \nabla_{2,1}^2 f(\boldsymbol{z}^*, \boldsymbol{D}) \odot \mathbb{1}_{S^*} \tag{43}$$

$$0 = R(\mathbf{J}^*, S^*) \tag{44}$$

### C.5 PROOF OF PROPOSITION 2.5 AND COROLLARY 2.6

**Proposition 2.5** *Let $N$ be the number of iterations and $K$ be the back-propagation depth. We assume that $\forall n \geq N-K$, $S^* \subset S_n$. Let $\bar{E}_N = S_n \setminus S^*$, let $L$ be the largest eigenvalue of $D_{:,S^*}^\top D_{:,S^*}$, and let $\mu_n$ be the smallest eigenvalue of $D_{:,S_n}^\top D_{:,S_{n-1}}$. Let $B_n = \left\| P_{\bar{E}_n} - D_{:,\bar{E}_n}^\top D_{:,S^*}^{\dagger\top} P_{S^*} \right\|$, where $P_S$ is the projection on $\mathbb{R}^S$ and $D^\dagger$ is the pseudo-inverse of $D$. We have*

$$\left\| J_l^N - J_l^* \right\| \leq \prod_{k=1}^K \left(1 - \frac{\mu_{N-k}}{L}\right) \|J_l^*\| + \frac{2}{L} \|D_l\| \sum_{k=0}^{K-1} \prod_{i=1}^k (1 - \frac{\mu_{N-i}}{L}) \left( \|z_l^{N-k} - z_l^*\| + B_{N-k} \|z_l^*\| \right) .$$

We denote by $G$ the matrix $(I - \frac{1}{L}D^\top D)$. For $z_N$ with support $S_N$ and $z*$ with support $S^*$, we have with the induction in Theorem 2.4

$$J_{l,S_N}^N = \left(G J_l^{N-1} + u_l^{N-1}\right)_{S_N} \tag{45}$$

$$J_{l,S^*}^* = \left(G J_l^* + u_l^*\right)_{S^*} \tag{46}$$

where $u_l^N = -\frac{1}{L}\left(D_l z_N^\top + (D_l^\top z_N - y_l)I\right)$ and the other terms on $\bar{S}_N$ and $\bar{S}^*$ are 0.
We can thus decompose their difference as the sum of two terms, one on the support $S^*$ and one on this complement $\bar{E}_N = S_N \setminus S^*$

$$J_l^* - J_l^N = (J_l^* - J_l^N)_{S^*} + (J_l^* - J_l^N)_{\bar{E}_N} .$$

Recall that we assume $S^* \subset S_N$. Let's study the terms separately on $S^*$ and $\bar{E}_N = S_N \setminus S^*$. These two terms can be decompose again to constitute a double recursion system,

$$(J_l^N - J_l^*)_{S^*} = G_{S^*}(J_l^{N-1} - J_l^*) + (u_l^{N-1} - u_l^*)_{S^*} \tag{47}$$

$$= G_{S^*,S^*}(J_l^{N-1} - J_l^*)_{S^*} + G_{S^*,\bar{E}_{N-1}}(J_l^{N-1} - J^*)_{\bar{E}_{N-1}} + (u_l^{N-1} - u_l^*)_{S^*} , \tag{48}$$

$$(J_l^N - J_l^*)_{\bar{E}_N} = (J_l^N)_{\bar{E}_N} = G_{\bar{E}_N}(J_l^{N-1} - J_l^*) + G_{\bar{E}_N,S^*}J_l^* + (u_l^{N-1})_{\bar{E}_N} \tag{49}$$

$$= G_{\bar{E}_N,S^*}(J_l^{N-1} - J_l^*)_{S^*} + G_{\bar{E}_N,\bar{E}_{N-1}}(J_l^{N-1} - J_l^*)_{\bar{E}_{N-1}} \tag{50}$$

$$+ (u_l^{N-1} - u_l^*)_{\bar{E}_N} + \left((u_l^*)_{\bar{E}_N} - D_{:,\bar{E}_N}^\top D_{:,S^*}(D_{:,S^*}^\top D_{:,S^*})^{-1}(u_l^*)_{S^*}\right) .$$

We define as $\mathcal{P}_{S_N,\bar{E}_N}$ the operator which projects a vector from $\bar{E}_N$ on $(S_N, \bar{E}_N)$ with zeros on $S_N$. As $S^* \cup \bar{E}_N = S_N$, we get by combining these two expressions,

$$(J_l^N - J_l^*)_{S_N} = G_{S_N,S_{N-1}}(J_l^{N-1} - J_l^*)_{S_{N-1}} + (u_l^{N-1} - u_l^*)_{S_N} \tag{51}$$

$$+ \mathcal{P}_{S_N,\bar{E}_N}\left((u_l^*)_{\bar{E}_N} - D_{:,\bar{E}_N}^\top D_{:,S^*}(D_{:,S^*}^\top D_{:,S^*})^{-1}(u_l^*)_{S^*}\right)$$

Taking the norm yields to the following inequality,

$$\left\| J_l^N - J_l^* \right\| \leq \left\| G_{S_N,S_{N-1}} \right\| \left\| J_l^{N-1} - J_l^* \right\| + \left\| u_l^{N-1} - u_l^* \right\| \tag{52}$$

$$+ \left\| (u_l^*)_{\bar{E}_N} - D_{:,\bar{E}_N}^\top D_{:,S^*}(D_{:,S^*}^\top D_{:,S^*})^{-1}(u_l^*)_{S^*} \right\| .$$

Denoting by $\mu_N$ the smallest eigenvalue of $D_{:,S_N}^\top D_{:,S_{N-1}}$, then $\left\| G_{S_N,S_{N-1}} \right\| = (1 - \frac{\mu_N}{L})$ and we get that

$$\left\| J_l^N - J_l^* \right\| \leq \prod_{k=1}^K (1 - \frac{\mu_{N-k}}{L}) \left\| J_l^{N-K} - J_l^* \right\| \tag{53}$$

$$+ \sum_{k=0}^{K-1} \prod_{i=1}^k (1 - \frac{\mu_{N-i}}{L}) \left( \left\| u_l^{N-k} - u_l^* \right\| + \left\| (u_l^*)_{\bar{E}_{N-k}} - D_{:,\bar{E}_{N-k}}^\top D_{:,S^*}^{\dagger\top}(u_l^*)_{S^*} \right\| \right) .$$

The back-propagation is initialized as $J_l^{N-K} = 0$. Therefore $\|J_l^{N-K} - J_l^*\| = \|J_l^*\|$. Moreover $\|u_l^{N-k} - u_l^*\| \le \frac{2}{L}\|D_l\|\|z_l^{N-k} - z_l^*\|$. Finally, $\left\|(u_l^*)_{\overline{E}_{N-k}} - D_{:,\overline{E}_{N-k}}^\top D_{:,S^*}^{\dagger\top}(u_l^*)_{S^*}\right\|$ can be rewritten with projection matrices $P_{\overline{E}_{N-k}}$ and $P_{\bar{S}^*}$ to obtain

$$\left\|(u_l^*)_{\overline{E}_{N-k}} - D_{:,\overline{E}_{N-k}}^\top D_{:,S^*}^{\dagger\top}(u_l^*)_{S^*}\right\| \le \left\|P_{\overline{E}_{N-k}}u_l^* - D_{:,\overline{E}_{N-k}}^\top D_{:,S^*}^{\dagger\top}P_{S^*}u_l^*\right\| \tag{54}$$

$$\le \left\|P_{\overline{E}_{N-k}} - D_{:,\overline{E}_{N-k}}^\top D_{:,S^*}^{\dagger\top}P_{S^*}\right\|\|u_l^*\| \tag{55}$$

$$\le \left\|P_{\overline{E}_{N-k}} - D_{:,\overline{E}_{N-k}}^\top D_{:,S^*}^{\dagger\top}P_{S^*}\right\|\frac{2}{L}\|D_l\|\|z_l^*\| \ . \tag{56}$$

Let $B_{N-k} = \left\|P_{\overline{E}_{N-k}} - D_{:,\overline{E}_{N-k}}^\top D_{:,S^*}^{\dagger\top}P_{S^*}\right\|$. We have

$$\|J_l^N - J_l^*\| \le \prod_{k=1}^{K}(1-\frac{\mu_{N-k}}{L})\|J_l^*\| + \frac{2}{L}\|D_l\|\sum_{k=0}^{K-1}\prod_{i=1}^{k}(1-\frac{\mu_{N-i}}{L})\Big(\|z_l^{N-k} - z_l^*\| + B_{N-k}\|z_l^*\|\Big) \ . \tag{57}$$

We now suppose that the support is reached at iteration $N - s$, with $s \ge K$. Therefore, $\forall n \in [N-s, N]$ $S_n = S^*$. Let $\Delta_n = F(z_n, D) - F(z^*, D) + \frac{L}{2}\|z_n - z^*\|$. On the support, $F$ is a $\mu$-strongly convex function and the convergence rate of $(z_N)$ is

$$\|z^* - z_N\| \le \big(1 - \frac{\mu}{L}\big)^s \frac{2\Delta_{N-s}}{L} \tag{58}$$

Thus, we obtain

$$\|J_l^N - J_l^*\| \le \prod_{k=1}^{K}(1 - \frac{\mu_{N-k}}{L})\|J_l^*\| \tag{59}$$

$$+ \frac{2}{L}\|D_l\|\sum_{k=0}^{K-1}\prod_{i=1}^{k}(1-\frac{\mu_{N-i}}{L})\Big(\|z_l^{N-k} - z_l^*\| + B_{N-k}\|u_l^*\|\Big)$$

$$\le \prod_{k=1}^{K}(1 - \frac{\mu_{N-k}}{L})\|J_l^*\| \tag{60}$$

$$+ \frac{2}{L}\|D_l\|\sum_{k=0}^{s-1}(1-\frac{\mu}{L})^k\Big(\|z_l^{N-k} - z_l^*\|\Big)$$

$$+ \frac{2}{L}\|D_l\|(1-\frac{\mu}{L})^s\sum_{k=s-1}^{K-1}\prod_{i=s-1}^{k}(1-\frac{\mu_{N-i}}{L})\Big(\|z_l^{N-k} - z_l^*\| + B_{N-k}\|(u_l^*)\|\Big)$$

$$\le \prod_{k=1}^{K}(1 - \frac{\mu_{N-k}}{L})\|J_l^*\| \tag{61}$$

$$+ \frac{2}{L}\|D_l\|\sum_{k=0}^{s-1}(1-\frac{\mu}{L})^k\big(1-\frac{\mu}{L}\big)^{s-1-k}\frac{2\Delta_{N-s}}{L}$$

$$+ \frac{2}{L}\|D_l\|(1-\frac{\mu}{L})^s\sum_{k=s-1}^{K-1}\prod_{i=s-1}^{k}(1-\frac{\mu_{N-i}}{L})\Big(\|z_l^{N-k} - z_l^*\| + B_{N-k}\|(u_l^*)\|\Big)$$

$$\le \prod_{k=1}^{K}(1 - \frac{\mu_{N-k}}{L})\|J_l^*\| \tag{62}$$

$$+ \|D_l\|(1-\frac{\mu}{L})^{s-1}s\frac{4\Delta_{N-s}}{L^2}$$

$$+ \frac{2}{L}\|D_l\|(1-\frac{\mu}{L})^s\sum_{k=s-1}^{K-1}\prod_{i=s-1}^{k}(1-\frac{\mu_{N-i}}{L})\Big(\|z_l^{N-k} - z_l^*\| + B_{N-k}\|(u_l^*)\|\Big)$$

$$\tag{63}$$

**Corollary 2.6** *Let $\mu > 0$ be the smallest eigenvalue of $D_{:,S^*}^\top D_{:,S^*}$. Let $K \leq N$ be the back-propagation depth and let $\Delta_N = F(z_N, D) - F(z^*, D) + \frac{L}{2}\|z_N - z^*\|$. Suppose that $\forall n \in [N - K, N]; S_n \subset S^*$. Then, we have*

$$\left\| J_l^* - J_l^N \right\| \leq \left(1 - \frac{\mu}{L}\right)^K \|J_l^*\| + K \left(1 - \frac{\mu}{L}\right)^{K-1} \|D_l\| \frac{4\Delta_{N-K}}{L^2} \ .$$

The term $\frac{2}{L}\|D_l\|(1 - \frac{\mu}{L})^s \sum_{k=s-1}^{K-1} \prod_{i=s-1}^{k}(1 - \frac{\mu_{N-i}}{L})\Big(\left\|z_l^{N-k} - z_l^*\right\| + B_{N-k}\|(u_l^*)\|\Big)$ vanishes when the algorithm is initialized on the support. Otherwise, it goes to 0 as $s, K \to N$ and $N \to \infty$ because $\forall n > N - s, \mu_n = \mu < 1$.

## D  ITERATIVE ALGORITHMS FOR SPARSE CODING RESOLUTION.

**ISTA.**  Algorithm to solve $\min_z \frac{1}{2}\|y - Dz\|_2^2 + \lambda\|z\|_1$

---
**Algorithm 1** ISTA
---
$y, D, \lambda, N$
$z_0 = 0, n = 0$
Compute the Lipschitz constant $L$ of $D^\top D$
**while** $n < N$ **do**
    $u_{n+1} \leftarrow z_N - \frac{1}{L}D^\top(Dz_n - y)$
    $z_{n+1} \leftarrow ST_{\frac{\lambda}{L}}(u_{n+1})$
    $n \leftarrow n + 1$
**end while**

---

**FISTA.**  Algorithm to solve $\min_z \frac{1}{2}\|y - Dz\|_2^2 + \lambda\|z\|_1$

---
**Algorithm 2** FISTA
---
$y, D, \lambda, N$
$z_0 = x_0 = 0, n = 0, t_0 = 1$
Compute the Lipschitz constant $L$ of $D^\top D$
**while** $n < N$ **do**
    $u_{n+1} \leftarrow z_n - \frac{1}{L}D^\top(Dz_n - y)$
    $x_{n+1} \leftarrow ST_{\frac{\lambda}{L}}(u_{n+1})$
    $t_{n+1} \leftarrow \frac{1+\sqrt{1+4t_n^2}}{2}$
    $z_{n+1} \leftarrow x_{n+1} + \frac{t_n-1}{t_{n+1}}(x_{n+1} - x_n)$
    $n \leftarrow n + 1$
**end while**

---

