# OpenReview forum: "Understanding approximate and unrolled dictionary learning for pattern recovery"
_ICLR.cc/2022/Conference — ICLR 2022 Poster_

### Official Review · Reviewer_njnY · 2021-10-20

**Correctness:** 2
**Technical Novelty And Significance:** 2
**Empirical Novelty And Significance:** 1
**Recommendation:** 3
**Confidence:** 4

**Main Review:**

*Strong points*

1. The technical writing of the paper is fairly well. The derivation can be followed without too much effort.

2. The paper presented an quite detailed discussion on unrolling scheme related to deep dictionary learning (DDL) which can be interested to the people working on DDL.

*Weak points*

1.  The applicability of the presented unrolling scheme is rather limited. It is only applicable to the problem motivated from a convex formulation of sparse coding, while a majority of the dictionary learning is using non-convex sparsity-regularization term such as -norm.

2. Its advantage over existing optimization methods e.g. K-SVD, proximal alternating method, and proximal alternating linearized method (PALM) for dictionary learning, is neither justified in theoretical analysis nor in experimental evaluation.

3. Experimental evaluation is limited with important missing information.

*Comments and Questions*


1. The hard part in dictionary learning is about the efficient treatment of non-convex constraints such as orthogonality or normalization. Are these treated using Lagrangian multiplier? if one want to call the proposed scheme for optimization model of dictionary learning

2. In Proposition 2.2 and 2.3. The results says there exist  such that the conclusion holds true. It seems that  is the cardinality of dictionary. Is this a typo, or they are the same.

3. While the proposed scheme is applicable to a specific learning scheme for MEG signal, it is not convincing enough to claim it is computationally efficient. The scale of such a problem is well handled by the traditional methods such as K-SVD, PAML.

4. In general, dictionary learning is not well-defined without additional constraints on the dictionary atoms, there is no mention of such constraints in the experiments.




**Summary Of The Paper:**

This paper present an efficient unrolling scheme for finding an approximate solution to the optimization problem arising from sparsity-based dictionary learning. Some asymptotically analysis on its gradient is presented and the method is also tested on analyzing MEG signal.

**Summary Of The Review:**

Numerical scheme for solving the non-convex problem related to dictionary learning has been extensively studied. For example, proximal alternating method and proximal alternating linearized method. The hard part indeed lies in efficient and effective estimation of dictionary with non-convex constraints. The proposed unrolling scheme does not address the true bottleneck of the problem. The asymptotic analysis presented indeed are rather the application of existing available techniques. Numerical experiments on MEG signals are also too limited to show its practical advantage

---

> ### Author Response · Authors · 2021-11-12
> **Thank you for your review**
>
> We thank the reviewer for pointing out that we forgot to mention we focus on unit norm constraints, where the atoms are normalized to avoid scaling issues. The main algorithm to solve this problem is to project the dictionary on the unit ball at each gradient step, independently of the method used to compute the gradient.
>
> > The applicability of the presented unrolling scheme is rather limited. It is only applicable to the problem motivated from a convex formulation of sparse coding, while a majority of the dictionary learning is using non-convex sparsity-regularization term such as -norm.
>
> While the study of non-convex constraints in dictionary learning is interesting, it is not the only subject of interest, especially in real-world applications where the Lasso is widely used. One can refer, for example, to Mairal et al. (2009), which is one of the most cited papers in dictionary learning. This problem is generally solved via alternating minimization, but this method fails to scale for MEG data.
> Moreover, the debate of “convex VS non-convex” approaches has existed for a while. A constructive discussion about $\ell_0$ vs $\ell_1$ can be found in the two following recent papers published in Statistical Science:
>
> Chen, Y., Taeb, A., & Bühlmann, P. (2020). A Look at Robustness and Stability of $\ell_ {1} $-versus $\ell_ {0} $-Regularization: Discussion of Papers by Bertsimas et al. and Hastie et al. Statistical Science, 35(4), 614-622.
>
> Hastie, T., Tibshirani, R., & Tibshirani, R. (2020). Best subset, forward stepwise or lasso? Analysis and recommendations based on extensive comparisons. Statistical Science, 35(4), 579-592.
>
> > In Proposition 2.2 and 2.3. The results says there exist such that the conclusion holds true. It seems that is the cardinality of dictionary. Is this a typo, or they are the same.
>
> It is indeed a typo. Thank you for noticing it. We will fix this notation in the revised manuscript.
>
> > While the proposed scheme is applicable to a specific learning scheme for MEG signal, it is not convincing enough to claim it is computationally efficient. The scale of such a problem is well handled by the traditional methods such as K-SVD, PAML.
>
> We respectfully but strongly disagree with the reviewer on this point. Traditional methods fail to scale when used for convolutional dictionary learning on dozens of minutes of recordings of MEG data. Convolutional dictionary learning packages that we could find on github (`sporco`, `alphacsc`) are mainly using the $\ell_1$ norm.
>
> > In general, dictionary learning is not well-defined without additional constraints on the dictionary atoms, there is no mention of such constraints in the experiments.
>
> We forgot to mention the unit norm constraint, where the atoms are normalized at each gradient step. We thank the reviewer for noticing this typo and we will modify the paper appropriately.
>
> > Numerical experiments on MEG signals are also too limited to show its practical advantage
>
> Our main point was to show that relevant patterns could be recovered faster with approximate dictionary learning than with state-of-the-art methods like `alphacsc`. We kindly ask the reviewer to give examples of experiments that would be interesting so that we can add them in the paper.

---

> > ### Comment · Reviewer_njnY · 2021-11-18
> > **some response**
> >
> > 1. The author's treatment on unit norm constraint for dictionary atom is a serious issue. For dictionary learning, the treatment of dictionary is more challenging than the treatment of sparse code. The unit form requirement on the atom of dictionary is one key challenge for its non-convexity, I don't see that a projection to a non-convex set is considered a sound step, as it leads to un-predictable behavior.   In addition,   the projection into a non-convex feasible set is not mentioned in the paper. The convergence analysis of the problem without the involvement of such a step becomes less meaningful.
> >
> > 2. The argument for $\ell_0$  norm over $\ell_1$ norm is not convincing. The author's argument is based on the linear regression problem, not for  non-linear problem as dictionary learning.  $\ell_1$ -norm and $\ell_0$-norm do  have their own advantages in  Lasso. The convexity of L1 norm makes its value by converting the whole formulation convex. For dictionary learning, L1-norm for sparse coding cannot make the problem convex, the whole problem remains non-convex. It is much less useful now to make one sub-iteration to be convex.

---

> > > ### Author Response · Authors · 2021-11-19
> > > **Answer to the reviewer**
> > >
> > > > 1. The author's treatment on unit norm constraint for dictionary atom is a serious issue. For dictionary learning, the treatment of dictionary is more challenging than the treatment of sparse code. The unit form requirement on the atom of dictionary is one key challenge for its non-convexity, I don't see that a projection to a non-convex set is considered a sound step, as it leads to un-predictable behavior. In addition, the projection into a non-convex feasible set is not mentioned in the paper. The convergence analysis of the problem without the involvement of such a step becomes less meaningful.
> > >
> > > We apologize if we were unclear regarding the constraint used in the dictionary. If the atoms belong to the unit ball, the problem is bi-convex (as stated in the revised version), while it is non-convex when they belong to the unit sphere.
> > >
> > > However, we first stress that our theoretical results do not depend on that constraint.
> > >
> > > Also, we strongly disagree with the statement: "I don't see that a projection to a non-convex set is considered a sound step, as it leads to unpredictable behavior". Indeed, an extensive literature on dictionary learning with both constraints is available, and several papers have led to important theoretical results.
> > >
> > > On the one hand, the unit-sphere constraint is studied in several important references in dictionary learning, including risk analysis (see for example [1,2,3,4]). We then kindly ask the reviewer for references that support the previous claim.
> > >
> > > On the other hand, several works use the unit-ball constraint, such as [5,6], among others.
> > > We could also cite other constraints such as the unit-ball/sphere on the Frobenius or Spectral norm of the dictionary [3,7]. In particular, the work in [7] shows only slight differences in practice between the unit-sphere and the unit-ball constraints, which we have also observed.
> > >
> > > In summary, we believe that the differences between all these constraints and their impact on dictionary learning are undoubtedly of great importance but out of the scope of this paper.
> > >
> > > [1] Gribonval, R., Jenatton, R., & Bach, F. (2015). Sparse and spurious: dictionary learning with noise and outliers. IEEE Transactions on Information Theory, 61(11), 6298-6319.
> > >
> > > [2] Mailhé, B., & Plumbley, M. D. (2012, March). Dictionary learning with large step gradient descent for sparse representations. In International Conference on Latent Variable Analysis and Signal Separation (pp. 231-238). Springer, Berlin, Heidelberg.
> > >
> > > [3] Kreutz-Delgado, K., & Rao, B. D. (2000, December). FOCUSS-based dictionary learning algorithms. In Wavelet Applications in Signal and Image Processing VIII (Vol. 4119, pp. 459-473). International Society for Optics and Photonics.
> > >
> > > [4] Jung, A., Eldar, Y. C., & Görtz, N. (2016). On the minimax risk of dictionary learning. IEEE Transactions on Information Theory, 62(3), 1501-1515.
> > >
> > > [5] Jenatton, R., Mairal, J., Obozinski, G., & Bach, F. R. (2010, January). Proximal methods for sparse hierarchical dictionary learning. In ICML.
> > >
> > > [6] Singh, S., Póczos, B., & Ma, J. (2018, March). Minimax reconstruction risk of convolutional sparse dictionary learning. In International Conference on Artificial Intelligence and Statistics (pp. 1327-1336). PMLR.
> > >
> > > [7] Yaghoobi, M., Blumensath, T., & Davies, M. E. (2009). Dictionary learning for sparse approximations with the majorization method. IEEE Transactions on Signal Processing, 57(6), 2178-2191.

---

> > > > ### Author Response · Authors · 2021-11-19
> > > > **End of the answer**
> > > >
> > > > > 2. The argument for $\ell_0$ norm over $\ell_1$ norm is not convincing. The author's argument is based on the linear regression problem, not for non-linear problem as dictionary learning. $\ell_1$ -norm and $\ell_0$-norm do have their own advantages in Lasso. The convexity of L1 norm makes its value by converting the whole formulation convex. For dictionary learning, L1-norm for sparse coding cannot make the problem convex, the whole problem remains non-convex. It is much less useful now to make one sub-iteration to be convex.
> > > >
> > > > The difference between $\ell_0$ and $\ell_1$ regularization is not limited to the convexity of the sparse coding problem, as the speed of convergence can largely differ depending on the algorithm chosen to solve the problem. Moreover, the given results on the linear regression problem cannot be ignored for Dictionary learning, as the sparse coding step is crucial. That being said, our point is not to debate the superiority of $\ell_0$ or $\ell_1$ norm. As you say, "$\ell_1$ -norm and $\ell_0$-norm do have their own advantages in Lasso" (the Lasso being the $\ell_1$ norm constrained problem, we understood that the reviewer means "sparse coding"). We have chosen the $\ell_1$ norm as a convenient framework for dictionary learning, as several authors (see references in the paper as well as the aforementioned papers)
> > > >
> > > > Finally, we kindly ask the reviewer to explain why the non-convexity constraint on the dictionary (cf. comment 1) "is a serious issue" while the sparse coding step should be chosen non-convex as "the whole problem remains non-convex." If we stick to this last argument, we could use the unit-sphere constraint without any problem, couldn’t we?

---

### Official Review · Reviewer_5JDj · 2021-10-30

**Correctness:** 3
**Technical Novelty And Significance:** 3
**Empirical Novelty And Significance:** 2
**Recommendation:** 6
**Confidence:** 5

**Main Review:**

Pros: The paper is written clearly. They study an interesting problem. Given the summary above, the stability analysis of the Jacobian and insights of limited-depth backpropagation to improve stability is new and of interest (proposition 2.5). The authors have provided comprehensive numerical results along with real applications to support their theories and claims. The visualization of loss landscape is interesting and has not previously explored for dictionary learning.

Cons: Given (Ablin et al., 2020, Tolooshams et al., 2020, and [1]), the novelty of the paper is incremental. Specifically, proposition 2.2 is known results from Ablin et al., 2020. Given the support and uniqueness assumption of the code, the reconstruction loss is strongly convex, and the optimization is reduced to the case in Ablin et al., 2020. Hence, proposition 2.3 follows. Stochastic CDL using unrolled networks has been previously explored in (Tolooshams et al., 2020) in the context of spike sorting and image denoising. Theoretical analysis and comparison of g1 and g2 for dictionary learning is known from [1].

The authors must clearly indicate their contributions and how their work differs from (Ablin et al., 2020, Tolooshams et al., 2020, and [1]). An additional analysis to make this work different from prior works can be providing dictionary recovery type of results similar to [2]. Overall, a better organization of the paper is recommended. See my detailed comments below.

- There are some statements that are not fully supported and are not completely accurate.
	- For example, In "Gradient estimation in dictionary learning" paragraph, the authors mention that " ... complete theoretical analysis of these problems are arduous". It is not clear what that means. Indeed, [2] is missing where it studies gradient-based alternating minimization for dictionary learning and provides convergence guarantees.
	- The statement on "... computing optimal sparse codes at each gradient step is unnecessary to recover the dictionary" is not precise. Indeed,  one factor that is crucial is sparsity of the code.
	- The statement before Section 3 is not generally true. For example, the instability issue does not exist for the case of strongly convex function or a another loss function.
	- The statement "we notice that there is no gain in the usage of DDL for the minimization of F_N without learning the steps sizes" seems to be contradicting the whole idea of "a better gradient estimation with DDL compared to AM". Elaborations needed.
	- The statement "... taking sub-windows of the full signal and performing SGD ..." is not precise. It depends whether you define a global loss function over the whole image or local for each patch.
	- The authors refer to the dictionary learning problem as non-convex. However, a more accurate statement is "bi-convex".

- How can you guarantee that invertibility of D^TD on the support stays along the whole iterations of alternating minimization?

- In dictionary learning, you usually learn the dictionary for a finite set of examples. However, (1) does not reflect this and aims to learn D for only one example y.

- The proof of proposition 2.2, the authors should clearly cite and highlight that Proposition 2.2 is a result from Ablin et al., 2020.

- How is lambda chosen to guarantee dictionary recovery in the numerical examples given the data generation of y = Dx? Given the overcomplete dictionary, very low lambda may result in false positive in support recovery which will affect the dictionary recovery result. The value of lambda is indeed crucial at the presence of noise to avoid noise overfitting.

- Experiments:
	- The paper has skipped a lot of experimental details in the main text and included only in the supp. Here are some that is recommended to be included in the main. How many iterations is "full"? At what iteration of the dictionary learning, Figure 1, 2, 3, 4 are based on (i.e., at dictionary initialization, after convergence, etc.)? Please elaborate what DL-Oracle is? How do you compute the optimal loss?
	- Why the dictionary error in Figure 4 (right) does not go to zero, and if their theory and analysis can explain that.
	- How to you define D of interest used in S(C)?
	- In Figure 5, do the authors train with various N values or train with the max N and look into the denoising performance given the code estimate at iteration n (i.e., Dx_n)?
	- What is the PSNR of the noisy image?
	- Intuitively, you may expect to find a trend in performance as you move from 100 batch size (which is not very low) to full batch. However, Figure 5 does not show such trend. Is there any insight for this?


- Other comments:
	- In the introduction, it is missed to mentioned that n  >> m.
	- "we propose to study the correlation between the gradients ...". It is not clear how we can show dictionary recovery given such comparison.
	- In (6), does y entails the entire data? Is there a notion of # of examples?
	- The change of y-axis from Figure 4 to 5 is confusing. Recommend to use the same x-axis on Figure 5 (left) as in Figure 4.
	- What is the SNR for when the noise variance is 0.1?
	- What are the properties of the dictionary and code, in addition to lambda, that can make support selection faster?
	- Recommend to differentiate between L in proposition 2.2 and 2.3 which are different.

- There are various typos and change of notation:
	- What is matrix A? there are numerous places that we see AD.
	- D has size of m x n but also n x L.
	- S is used once for the support and another time as a matric for dictionary error.
	- In the proof of Proposition 2.1, argmin must be over D \in C not D in general. In addition, z must have dimension m not L.

[1] B. Tolooshams and D. Ba, "PUDLE: implicit acceleration of dictionary learning by backpropagations", 2021.

[2] N. Chatterji and P. Bartlett, "Alternating minimization for dictionary learning: Local convergence guarantees", 2017.


------------
after discussion

See discussion on "Revision of the paper" comment initiated by the author's for my updated review. I hope the authors find my overall comments and the discussion useful in improving their manuscript. My main concern had been in relation to prior work and its novelty. The authors have addressed this with proper citation and additional explanation in the intro. I have updated my rating accordingly, increased the technical novelty to 3 (given the focus of the paper on the regime before support identification), however still kept the empirical novelty to 2.

**Summary Of The Paper:**

The paper studies dictionary learning where assumes that data can be represented as a linear combination of a few atoms of a matrix called dictionary. Traditionally, one way to approach the problem is to set up a min-min (bi-convex) optimization problem known as lasso or basis pursuit and solve it through alternating minimization (alternate between a sparse coding step and a dictionary update). The paper compares gradient based alternating minimization which uses an analytic gradient (given the code estimate, compute the gradient) to unrolled-based dictionary learning which uses backpropagation (automatic differentiation) through an iterative algorithm estimating the code (inner problem) to compute the gradient for update of the dictionary. This paper borrows results from (Ablin et al., 2020) that had studied min-min optimization problems when the objective functions are smooth, differentiable, and strongly convex. Specifically, solving lasso iteratively through ISTA or FISTA, after support selection, with some assumption on the dictionary, the problem of this paper is reduced to the strongly convex case of (Ablin et al., 2020). Hence, the results follow. Their contribution that makes their paper different from (Ablin et al., 2020) is the study of the Jacobian and instability of the convergence prior to support selection. This very same model (dictionary learning through unrolled algorithms) have been already studied theoretically by [1] (which is missing in the citations) and empirically in the context of convolutional dictionary learning by this paper (Tolooshams et al., 2020) that they cite.

**Summary Of The Review:**

The paper contains some new results compared to prior works (stability analysis of the Jacobian and limited-depth backprop). However, the papers lacks novelty given the works from Ablin et al., 2020, Tolooshams et al., 2020, and [1] (see above for details). Given the current formulation, results, and version of the paper, the paper's contributions do not distinguish itself from prior works. Hence, I do not recommend an acceptance of this paper. I hope that authors find my comments helpful and can address my concerns detailed in the review.

------------
after discussion

The authors explained how their work with focus on instability of the Jacobian and analysis prior to support identification distinguishes itself from prior studies. Given this and the additional citations, I have increased my score.

---

> ### Author Response · Authors · 2021-11-12
> **Thank you for your review**
>
> The main remark of the reviewer is that this work is only incremental and not very different from Ablin et al. (2020), Tolooshams et al. (2020), and Tolooshams and Ba (2021) and that we do not give enough credit to prior work. We respectfully disagree with the reviewer on this point for the following reasons:
>
> Ablin et al. (2020) study smooth losses, whereas we study the Lasso and extend their results to the non-smooth case in dictionary learning. We make mention of their work multiple times in the paper as well as in the proofs. More specifically, you mention that our results follow Ablin et al. (2020). However, we must stress that this is only true on the support. As far as we know, the computation of the Jacobian and the study of numerical instabilities in the non-smooth case have never been studied and are of critical interest to better understand unrolling for dictionary learning.
>
> We mention Tolooshams et al. (2020) in the introduction, and although their work is relevant in the context of the paper, it does not address the same practical considerations at all. Indeed, it is an EM based algorithm designed to choose the optimal value of lambda, and their work does not give any practical details on the proper choice of number of iterations in unrolling. Moreover, they do not provide a comparison to AM (unrolling without backpropagation).
>
> Tolooshams and Ba (2021) study unrolled DL in a regime where the support is recovered. On the contrary, we take a particular interest in what happens outside of the support in our contribution. This appears to be of critical practical interest when using unrolling, especially with few iterations. Although Tolooshams and Ba (2021) is clearly related to our paper, their framework does not apply to unrolling and AM in early iterations regimes. Finally, we will, of course, cite Tolooshams and Ba (2021), but we must stress that this paper is still under review, and that the preprint was released on arxiv at the end of May 2021, so we couldn’t have been aware of this work when we started ours. We then respectfully disagree when the reviewer claims that our contribution is only incremental compared to their work.
>
> We provide answers to the reviewer’s detailed comments below.
>
> > For example, In "Gradient estimation in dictionary learning" paragraph, the authors mention that " ... complete theoretical analysis of these problems are arduous". It is not clear what that means. Indeed, [2] is missing where it studies gradient-based alternating minimization for dictionary learning and provides convergence guarantees.
>
> As far as we know, the study of non-convex problems is complex, and there is no algorithm to find global optima in a reasonable time. This is what we meant by “complete theoretical analysis of these problems are arduous.”
>
> > The statement on "... computing optimal sparse codes at each gradient step is unnecessary to recover the dictionary" is not precise. Indeed, one factor that is crucial is sparsity of the code.
>
> The point of the paper is to see if it is relevant to compute the gradient steps with a small number of iterations to recover patterns in a signal, e.g., the dictionary, without taking into account the reconstruction quality.
>
> > The statement before Section 3 is not generally true. For example, the instability issue does not exist for the case of strongly convex function or a another loss function.
>
> While we agree with the fact that “the instability issue does not exist for the case of strongly convex function or a another loss function.”, we clearly stated at the beginning of the paper that we study the Lasso and stick to this case. Our manuscript is not meant to be a review paper of all existing losses for dictionary learning.
>
> > The statement "we notice that there is no gain in the usage of DDL for the minimization of F_N without learning the steps sizes" seems to be contradicting the whole idea of "a better gradient estimation with DDL compared to AM". Elaborations needed.
>
> The complete sentence is “When looking at the loss and the recovery score, we notice that there is no gain in the usage of DDL for the minimization of F_N without learning the steps sizes, but there is an increase of performance concerning the recovery score.”, which is not in contradiction to the general idea of the paper. As we already mentioned, we focus on the dictionary recovery.
>
> > The authors refer to the dictionary learning problem as non-convex. However, a more accurate statement is "bi-convex".
>
> If the usual general formulation is indeed “bi-convex,” and we will precise it, we must stress that the approximate problem we are studying is not convex in D.

---

> > ### Author Response · Authors · 2021-11-12
> > **Answer to the last comments**
> >
> > > How can you guarantee that invertibility of $D^TD$ on the support stays along the whole iterations of alternating minimization?
> >
> > As mentioned in the manuscript, we assume that $D^TD$ is invertible on the support and we do not claim to have such guarantee. However, results by Tibshirani 2013 says that for dictionaries $D$ drawn from continuous values distributions, this is the case with probability 1. Besides, we focus on what is going on outside of the support in the paper, and guaranteeing the invertibility of D^TD on the support is not central in practice with a small number of iterations.
> >
> > > In dictionary learning, you usually learn the dictionary for a finite set of examples. However, (1) does not reflect this and aims to learn D for only one example y.
> >
> > Thank you for this remark. We will change our notations so that it is clear we learn from multiple examples.
> >
> > > The proof of proposition 2.2, the authors should clearly cite and highlight that Proposition 2.2 is a result from Ablin et al., 2020.
> >
> > Proposition.1 in Ablin et al. 2020 is simply obtained using the definition of the smoothness of the outer loss (which is directly supposed in their assumption). Proposition 2.2 gives a similar result but based on the fact that F is locally lipschitz for any $D$. This result, while similar, is not a direct application of the result by Ablin et al. (2020). We will however better link the two in our revised version.
> >
> > > How is lambda chosen to guarantee dictionary recovery in the numerical examples given the data generation of $y = Dx$? Given the overcomplete dictionary, very low lambda may result in false positive in support recovery which will affect the dictionary recovery result. The value of lambda is indeed crucial at the presence of noise to avoid noise overfitting.
> >
> > We agree that the value of lambda is crucial. However, we don’t aim to improve the choice of hyperparameters in our work. See for example Tolooshams et al. (2020) for an algorithm to choose an appropriate value of lambda.
> >
> > > The paper has skipped a lot of experimental details in the main text and included only in the supp. Here are some that is recommended to be included in the main. How many iterations is "full"? At what iteration of the dictionary learning, Figure 1, 2, 3, 4 are based on (i.e., at dictionary initialization, after convergence, etc.)? Please elaborate what DL-Oracle is? How do you compute the optimal loss?
> >
> > Because of the limited space, we did not add all experimental details in the main text. However, we fully described our setups in the appendix. We will try to add as much information as possible if the reviewer thinks it is relevant. To answer the reviewer questions, “full” refers to the full backpropagation, as described in the caption of the figure, so as many backpropagation steps as there was iteration in the forward. Figure 1, 2, 3 correspond to random gaussian dictionaries and Figure 4 is evaluated after convergence. DL-Oracle corresponds to the performances obtained by solving each of the inner problems up to convergence. The optimal loss is computed using such DL-Oracle.
> >
> > > Why the dictionary error in Figure 4 (right) does not go to zero, and if their theory and analysis can explain that.
> >
> > Lasso-based DL does not guarantee to recover the dictionary that has generated the data, especially in the presence of noise. This is why the optimal dictionary recovery with full AM may be different from the ground truth.
> >
> > > How to you define D of interest used in S(C)?
> >
> > The dictionary of interest is the dictionary used to generate the data in synthetic experiments. For real experiments, we compare to the dictionary found by complete AM, even though we have no guarantee that it is the best dictionary. We will make it clearer in the manuscript.
> >
> > > In Figure 5, do the authors train with various N values or train with the max N and look into the denoising performance given the code estimate at iteration n (i.e., Dx_n)?
> >
> > In Figure 5, we train and evaluate using only N iterations of sparse coding. We will clarify this point in the paper.
> >
> > > Intuitively, you may expect to find a trend in performance as you move from 100 batch size (which is not very low) to full batch. However, Figure 5 does not show such trend. Is there any insight for this?
> >
> > As the size of the batch impacts the computational time, there is a tradeoff between how precise the gradient estimate is and how long it takes to compute it. This explains why there is no such trend. Moreover, as we rely on a stochastic line search method which is more noisy than full batch line search, this also explains why full batch has a better convergence behavior than stochastic line search with big batches.
> >
> > > Other comments and typos:
> >
> > Thank you for these comments, we will include all missing information in the paper and we will correct the typos. Again, we do not focus on support selection issues, as we study unrolling and AM in early iterations regimes.

---

### Official Review · Reviewer_S6uo · 2021-11-02

**Correctness:** 3
**Technical Novelty And Significance:** 2
**Empirical Novelty And Significance:** 2
**Recommendation:** 6
**Confidence:** 4

**Main Review:**

Pros:
- The topic is important and timely.
- Unrolling optimization algorithms and coupling them via backpropagation is a natural direction being pursued in several application domains. It is nice to see some careful analysis for this specific context (dictionary learning).
- The paper is nicely and clearly written.
- The experiments are thorough and well conducted.

Cons:
- The significance of the proposed method (and analysis), and overall relative contributions to the state of the art, is somewhat unclear.
- Theoretical analysis of dictionary learning is a well-studied problem (with all kinds of running time and sample complexity guarantees).
It will be helpful to precisely position the quantitative guarantees (Theorem 2.4, Prop 2.6) in the literature whether there is any particular benefit or drawback of computing the (outer) sub-gradient over, say, AM or other techniques.
- Backpropagation through the solution of sparse recovery has been already done before (by Bertrand et al, ICML 2020), so a clear comparison with this previous approach (and discussions of pros vs cons of backpropagation through unrolling) will be beneficial.
- The experimental results in Figs 3/4 don't really show significant improvement over AM. It may be helpful to illustrate what lessons can be learned here.
- Since the MEG application addresses convolutional DL, it may be helpful to address whether variations of the analysis approach is applicable to the convolutional case.

---

Thanks for your response and for clarifying my questions regarding novelty. Bumping my score up to 6.

**Summary Of The Paper:**

The authors theoretically study the performance of dictionary learning using "unrolling" based methods. As opposed to alternating minimization (AM) which switches back and forth between dictionary estimation and sparse recovery, the paper writes down the target dictionary as the solution to a bi-level optimization, where the "inner" optimization is approximated by unrolling with N steps. The main contribution is an approach (along with careful analysis) of computing the subgradient for the outer optimization, and experiments to show that this method works on synthetic and real datasets.

**Summary Of The Review:**

This nicely written paper theoretically studies the dynamics of dictionary learning via unrolling + backpropagation style training. However, the authors could consider contrasting the results with the existing literature and clearly articulating how/where unrolling-style training is better or worse than the current of the art.

---

> ### Author Response · Authors · 2021-11-12
> **Thank you for your review**
>
> Thank you for your constructive feedback and for appreciating the clarity of our manuscript and its relevance to the dictionary learning community. From our understanding, the main weakness that the review highlights in our paper is the clarity of our contributions and their position relative to other theoretical analyses of dictionary learning.
>
> Our main goal is to study non-smooth unrolling in a setting where the support has not been identified. Up to our knowledge, this analysis has not been done yet, even though it is of critical practical interest. Moreover, we focus on dictionary recovery instead of reconstruction quality. We study whether it is beneficial to learn the dictionary with iterates that have not converged to improve the computational efficiency. We compare AM and unrolling in an early iterations regime to see which leads to the more acceptable results.
>
> Our main contribution is Proposition 2.5, which shows that there is an intrinsic instability in the unrolling. The point of our theoretical study was to emphasize that there is no convergence guarantee outside of the support and that using unrolling with too many iterations leads to unstable gradient estimates. Bertrand et al. (2020) study hyperparameter optimization in the LASSO with full convergence of the iterates. The authors compute the jacobian of the hyperparameter and introduce a first-order optimization method to find a good fit with respect to the data and a different loss. Here, the knowledge of the jacobian is useless when the iterates have converged, as the gradient has a closed-form solution in this case. While the theory behind their results is based on similar techniques, as mentioned in the paper,  the framework remains very different, making the comparison difficult.
>
> Then, we empirically compare unrolling, and partial AM (stopped at a fixed iteration) in section 3.1, where one can see that the dictionary recovery is better with unrolling for a limited number of iterations only, and those good dictionaries are recovered even with imprecise sparse codes.
>
> The main lessons to be learned from our work are the following:
> 1. Unrolling leads to better results than AM (with an equal number of sparse coding iterations) only for a small number of iterations.
> 2. It is possible to learn a good dictionary with a small number of iterations.
>
> > Since the MEG application addresses convolutional DL, it may be helpful to address whether variations of the analysis approach is applicable to the convolutional case.
>
> In the paper, we will clarify that our analysis also applies to convolutions, which are still linear operators.

---

> > ### Comment · Reviewer_S6uo · 2021-11-30
> > **Thanks**
> >
> > for your response and for answering my questions regarding the surrounding literature.

---

### Official Review · Reviewer_EyuW · 2021-11-05

**Correctness:** 4
**Technical Novelty And Significance:** 3
**Empirical Novelty And Significance:** 3
**Recommendation:** 8
**Confidence:** 3

**Main Review:**

From my point of view, this paper is well-written and the presented analysis is interesting in the context of a recently growing body of work on unrolling. It provides a rigorous theoretical investigation and justification of practically observable phenomena.

I only have one minor point which can hopefully be clarified. Maybe the authors can comment on the role of $L$ in Proposition 2.2. Is it correct that $L$ is at the same time the row dimension of $D$ and a constant to estimate convergence speed?

**Summary Of The Paper:**

The authors investigate the asymptotic behavior of unrolling applied to dictionary learning. The applicability of unrolling to dictionary learning results from the circumstance that dictionary learning can be reformulated in terms of bilevel optimization, where the lower-level (or inner) problem is a sparse coding problem (in the LASSO case considered here). Unrolling means to replace the argmin in the lower-level problem with the $N$-th iterate of a suitable optimization algorithm. Gradients of the upper-level loss can then be computed by means of backpropagation through algorithmic iterates. The authors study the behavior of resulting gradients depending on $N$ and draw a comparison with alternating minimization. They find that unrolling constitutes a scalable alternative to alternating minimization, where unrolling a relatively small number of iterations or using truncated backpropagation is favorable to ensure stable approximate gradients.

**Summary Of The Review:**

I think that this paper should be accepted. As far as I can see, the presented results are relevant and correct. However, I did not check the proofs in the last detail.

---

> ### Author Response · Authors · 2021-11-12
> **Thank you for your review**
>
> Thank you for your very positive feedback.
>
> > Maybe the authors can comment on the role of L in Proposition 2.2. Is it correct that L is at the same time the row dimension of D and a constant to estimate convergence speed?
>
> Thank you for noticing this typo. Indeed, there is no link between the dimension L and the constant in the convergence bound. We will modify these notations in our revision.

---

### Author Response · Authors · 2021-11-17
**Revision of the paper**

We thank the reviewers for their insightful feedback. We uploaded a revised version of the paper with the following modifications (major changes are highlighted in red in the revised manuscript):
- We corrected typos, including the dimension of D and z, and we made it clear that we train on several samples.
- We added references to Tolooshams and Ba (2021).
- We emphasized that our results apply for Lasso-based dictionary learning only, and we mentioned that we use the unit norm constraint.
- We added experimental details in the figures
- We mentioned that dictionary learning is bi-convex.
- We mentioned that our theoretical and empirical results also apply to convolutions.

We completely rewrote the paragraph explaining our contributions in the introduction (last paragraph before section 2) to clarify the differences with Tolooshams and Ba 2021, Bertrand et al. 2020, and Ablin et al. 2020, as it was indeed not clear enough in the prior version. Again, we would like to recall the main contributions of our theoretical and empirical study on unrolled dictionary learning:
- The estimation of the Jacobian in lasso-based dictionary learning is unstable before reaching the support, which makes unrolled sparse coding inefficient after a few dozens of iterations.
- Unrolling leads to better results than AM (with an equal number of sparse coding iterations, i.e., Iterations N) only for a small number of iterations.
- It is possible to learn a good dictionary with a small number of iterations. We include a proof of concept on MEG data.

We highlighted that "Iterations N" in Figures 1, 2, 3, 4 stand for unrolled sparse coding iterations, as this may not have been clear enough for several reviewers. Indeed, the paper's goal is to compare AM and unrolling depending on the number of sparse coding iterations -- or equivalently layers.

Finally, we implemented K-SVD for convolutional dictionary learning by adapting Yellin et al. (2017) to our rank-1 constraint (Equation 6) to highlight its scaling issue in our context. First, note that this is a non-trivial implementation as K-SVD with rank-1 constraint does not exist. Moreover, to make it runnable, we had to implement efficient correlation updates in the sparse coding phase (see l.150 in `ksvd_cdl.py`), which we could not find described in the literature. We were not able to make it run in less than 2h on our problem because of the cost of SVD. In our original experiment, we are considering a signal with $203$ channels and $41,657$ time samples (sampling frequency of $150Hz$), and we are looking for $40$ atoms of length 1s. In this case, the residual matrices on which we perform the SVD are huge: the number of activations for the atom $k$ $s_k \approxeq 1000$ times the shape of an atom $150\times300 = 45 000$. This makes the algorithm very slow and requires a very large memory to compute the result of the SVD. We were able to run a smaller case experiment with only 20 atoms of length 0.5s (75 time stamps) in 6693s on a large machine with 72 cores and 300Gb of RAM - larger than the one used for our experiments. Note that the SVD makes use of around 30 cores as it is using the NumPy implementation with openBLAS primitives. We put our code in the supplementary materials as well as output files `atoms_ksvd.pdf` and `objective_evolution.pdf.` We hope that this experiment will convince the reviewers that the K-SVD algorithm does not scale well on our data.

---

> ### Comment · Reviewer_5JDj · 2021-11-18
> **Comments on author's response and revision**
>
> I thank the authors for the response to each of the reviewer's comments and also the detailed list of modifications made to the paper.
>
> I agree with the authors and again acknowledge (as I also pointed out on Pros of my review) that the analysis on the Jacobian stability and limited-depth backpropagation due to their analysis on instability of the gradient from early iterations (prior to support identification) is new in this paper. Regarding the novelty and incremental nature of the work, I would like to clear out how I see the relation of this work to prior work.
>
> - This paper vs. Tolooshams et al. (2020): The focus of this paper is different from Tolooshams et al. (2020). However, the MEG application using stochastic CDL is not a novel experiment as similar network is used in Tolooshams et al. (2020) for CDL in the context of spike sorting (i.e., a neural analysis on extracellular voltage recording data from the brain to find some localized patterns). Hence, the paper must clearly state that the network that they call DDL is previously proposed in the literature and used in various applications. In this paper as the authors mentioned, they focus on the theoretical analysis of DDL and its Jacobian/gradient instability.
>
> - This paper vs. Ablin et al. (2020) and Tolooshams and Ba (2021): As I mentioned before, the paper must clearly state which of the results are covered by the afore-mentioned prior works or the fact that Tolooshams and Ba (2021) has studied the very same problem but with the focus on support identification.
>
> Given the authors' response and updated paper and the comments from other reviewers,
>
> 1. The authors have included in the paper how their work differs from prior works (Bertrand et al. (2020), Ablin et al. (2020), Tolooshams et al. (2020), Tolooshams and Ba (2021)). This is a positive point.
>
> 2. I found the author's response regarding $\ell_0$ and $\ell_1$ to reviewer njnY convincing. I point out that $\ell_1$-norm based dictionary learning/sparse coding with Lasso has become widely accepted and popular in the computer science community and has been studied in the literature. So, in my opinion, studying dictionary learning with $\ell_1$-norm is the choice of author to focus on. In the absence of $\ell_0$ analysis, it is still of interest to the community.
>
> 3. Given the K-SVD computational issues, I recommend the authors to compare their method to Scetbon, 2019 "Deep K-SVD Denoising" if they see related.

---

> > ### Author Response · Authors · 2021-11-19
> > **Thank you for your comment**
> >
> > > I agree with the authors and again acknowledge (as I also pointed out on Pros of my review) that the analysis on the Jacobian stability and limited-depth backpropagation due to their analysis on instability of the gradient from early iterations (prior to support identification) is new in this paper. Regarding the novelty and incremental nature of the work, I would like to clear out how I see the relation of this work to prior work.
> >
> > Thank you for acknowledging the novelty regarding the analysis on the Jacobian stability and the behavior in early iterations. This analysis makes us believe that this work is not incremental, especially regarding the intensive experimental study which supports this theoretical analysis.
> >
> > > This paper vs. Tolooshams et al. (2020): The focus of this paper is different from Tolooshams et al. (2020). However, the MEG application using stochastic CDL is not a novel experiment as similar network is used in Tolooshams et al. (2020) for CDL in the context of spike sorting (i.e., a neural analysis on extracellular voltage recording data from the brain to find some localized patterns). Hence, the paper must clearly state that the network that they call DDL is previously proposed in the literature and used in various applications. In this paper as the authors mentioned, they focus on the theoretical analysis of DDL and its Jacobian/gradient instability.
> >
> > We fully agree with the reviewer on the fact that DDL has already been proposed in various works. We uploaded a slightly modified version of the introduction of our paper to make this clearer. We hope it is sufficient to properly state that we aim to study an algorithm which already exists. Regarding the application, our point was to demonstrate that DDL scales to large MEG multivariate time series by adapting particular constraints introduced in Dupré La Tour et al. (2018) to unrolling. Indeed, these constraints lead to impressive results on MEG data, and we wanted to highlight our manuscript message which is that unrolling with a few iterations could obtain similar performances for atom recovery compared to alphacsc (Dupré La Tour et al. (2018)) with improved computational efficiency. While the stochastic algorithm (Vaswani et al. (2019)) and the settings we use are different from Tolooshams et al. (2020) (we explain why this particular algorithm is helpful in the context of dictionary learning in the paper), we acknowledge that Tolooshams et al. (2020) already applied a stochastic CDL algorithm to medical data. We modified the paper accordingly (end of section 3.2).
> >
> > > This paper vs. Ablin et al. (2020) and Tolooshams and Ba (2021): As I mentioned before, the paper must clearly state which of the results are covered by the afore-mentioned prior works or the fact that Tolooshams and Ba (2021) has studied the very same problem but with the focus on support identification.
> >
> > We uploaded another version of the paper where we modified the paragraph following Proposition 2.3 to better explain the results by Ablin et al. (2020) and Tolooshams and Ba (2021), including the fact that they studied this problem with the focus on support identification. We also related Corollary 2.6 to results in Tolooshams and Ba (2021). We hope this gives better credit to their works.

---

> > > ### Author Response · Authors · 2021-11-19
> > > **End of the answer**
> > >
> > > > Given the authors' response and updated paper and the comments from other reviewers,
> > > The authors have included in the paper how their work differs from prior works (Bertrand et al. (2020), Ablin et al. (2020), Tolooshams et al. (2020), Tolooshams and Ba (2021)). This is a positive point.
> > >
> > > Thank you for this positive feedback
> > >
> > > > I found the author's response regarding $\ell_0$ and $\ell_1$ to reviewer njnY convincing. I point out that $\ell_1$-norm based dictionary learning/sparse coding with Lasso has become widely accepted and popular in the computer science community and has been studied in the literature. So, in my opinion, studying dictionary learning with $\ell_1$-norm is the choice of author to focus on. In the absence of $\ell_0$ analysis, it is still of interest to the community.
> > >
> > > Thank you for this comment. We fully agree.
> > >
> > > > Given the K-SVD computational issues, I recommend the authors to compare their method to Scetbon, 2019 "Deep K-SVD Denoising" if they see related.
> > >
> > > Regarding Reviewer njnY’s comment, we think it would be interesting to compare various updates/constraints on the dictionary and the sparse coding step. However, we would like to point out that the Deep K-SVD is also using $\ell_1$ formulation (see [eq (7) and above in the paper](https://arxiv.org/pdf/1909.13164.pdf)). The model is very similar to DDL, with an extra network to select the hyper-parameter $\lambda$.  Moreover, their focus is on achieving good denoising performances with supervised training and not on recovering a dictionary. We thus believe such a comparison would not be relevant for our paper.
> > >
> > > Given that we have addressed the remarks raised in the review and that the reviewer deems that our paper provides novel and interesting results, we kindly ask him/her to revise his/her rating of our work.

---

### Decision · Program_Chairs · 2022-01-20

**Decision:**

Accept (Poster)

**Comment:**

The paper proposes an unrolled algorithm to solve the l1-norm formulated dictionary learning problem, and focuses on the number of unrolling steps. It shows that it is better to limit the number of unrolling steps, and this leads to favorable performance over the alternating minimization baseline. The method can also be adapted to scale to very large datasets.

Most reviewers were positive or became positive after the rebuttals.  Reviewer njnY was still concerned about some issues, such as constraints and the choice of the l1 model over the l0 model; there also may have been confusion about unit sphere vs unit ball constraints.  However, given the recommendations of the other reviewers and my own opinion, I think the paper is a worthy contribution, and the point about not unrolling too deeply is an important one that is worth highlighting.